# Response Mechanisms of Adventitious Root Architectural Characteristics of *Nitraria tangutorum* Shrubs to Soil Nutrients in Nabkha

**DOI:** 10.3390/plants11233218

**Published:** 2022-11-24

**Authors:** Xiaole Li, Xiaohong Dang, Yong Gao, Zhongju Meng, Xue Chen, Yanyi Wang

**Affiliations:** 1College of Desert Control Science and Engineering, Inner Mongolia Agricultural University, Hohhot 010018, China; 2Inner Mongolia Hangjin Desert Ecosystem National Positional Research Station, Ordos 017400, China

**Keywords:** adventitious roots, root architecture, soil nutrients

## Abstract

The adventitious roots of desert shrubs respond to a nabkhas soil environment by adjusting their configuration characteristics, but the mechanism of this response and the main influencing factors are still unclear. To illustrate this response pattern, *Nitraria tangutorum* Bobrov, Sovetsk. in West Ordos National Nature Reserve was studied, and the shrub was divided into three growth stages: the rudimental stage, developing stage, and stabilizing stage. A combination of total root excavation and root tracing was used to investigate their adventitious root morphology. The results show the following: (1) As the shrub grows, the ability to accumulate sand into nabkhas increases. (2) The soil nutrient accumulation capacity increased with shrub growth. The “fertilizer island effect” was formed in the nutrient developing stage and stabilizing stage of nabkhas soil, but the rudimental stage was not formed. (3) The adventitious root architecture of *N. tangutorum* at different growth stages was all herringbone with a simple branch structure. With the growth in *N. tangutorum*, the root diameter of each level gradually increased, the branches of the shrub grew gradually complicated, and the range of resource utilization gradually expanded. (4) Redundancy analysis (RDA) results show that soil organic carbon (SOC) was the main factor affecting the adventitious root architecture. The results of this study reveal the adjustments the adventitious root architecture of *N. tangutorum* make in order to adapt to the stress environment and provide data support for the protection of natural vegetation in West Ordos.

## 1. Introduction

Nabkha is a geomorphological form, formed by the accumulation of wind and sand flow encountering shrub blockage in arid and semi-arid desert areas [1]. Nabkha contains a large amount of information on regional environmental changes, which can comprehensively reflect the dynamic environment and material sources in the deposition area and provide an important parameter indicator to describe the regional soil wind erosion and land degradation status [2]. However, shrub influence, through interception, root secretion, and apoplankton, creates a “fertilizer island effect”, in which the nutrients of nabkha soils are higher than those without vegetation cover [3]. This can better promote plant growth to achieve the effect of fixing drifting sand [4]. Therefore, shrub has been a hot research topic in recent years. The formation of nabkhas depends on multiple influences, such as sand source, wind, and plants, among which vegetation is a key factor affecting the morphology of nabkhas [5]. It is generally agreed that the plant wind-blocking benefits depend on the plant sidelight area on the windward side [6,7], and taller and denser shrubs are more likely to form nabkhas due to their stronger interception capacity [8,9]. However, some of the shrubs cannot form larger nabkhas, although they do possess the ability to accumulate sand into dunes. It has been shown that the formation of nabkhas is strongly related to their adventitious roots, and plants that can produce adventitious roots after being buried in sand are more likely to form tall nabkhas [10]. Therefore, it is important to explore adventitious roots to further understand nabkhas.

The root system is an important organ for material exchanges between plants and the external environment [11]. Root system conformation is the distribution and arrangement of the root system in the soil [12]. To adapt to the environment to the greatest extent possible, the root architecture of plants dynamically adjusts to changes in the soil environment [13]. A good root architecture facilitates plants’ ability to improve the root’s morphological stability and resource utilization efficiency during adversity stress [11]. This is a feedback mechanism and part of the plants’ ecological adaptation strategy to heterogeneous soil resources [14]. It has been found that different plants exhibit different root adaptation strategies in similar environments [15]. Plants at different growth stages adapt to changing environments by adjusting their root configuration [16]. What kind of adaptation strategies provide an adventitious root conformation under the influence of the nabkha soil environment exhibit?

There is a lack of research on the adventitious roots’ strategies to adapt to stressful environments during growth in desert shrub. It is also unclear how the nabkha soil environment affects adventitious root configuration. Therefore, our experiment aimed to reveal the changes in the “fertilizer island effect” caused by shrub growth in the same bioclimatic zone. In this study, the response of the root morphology to changes in the soil environment under the influence of the “fertilizer island effect” and the main influencing factors are identified, and we provide a theoretical basis and data support for the conservation and restoration of desert plants.

## 2. Results

### 2.1. Morphological Characteristics of N. tangutorum Shrub

The nabkhas of *N. tangutorum* in West Ordos National Nature Reserve are mostly semi-ellipsoid or shield shaped. As shown in Table 1, the morphological parameters of *N. tangutorum* were W_s_ 2.32–4.32 m, L_s_ 2.87–5.50 m, H_s_ 0.86–1.54 m, and C 2.81–9.73 m^2^. The morphological parameters were positively correlated with the growth and development of *N. tangutorum*. The mean morphological parameters of nabkhas were as follows: W_n_ 2.28–5.23 m, L_n_ 2.74–6.17 m, H_n_ 0.46–0.91 m, S_n_ 4.91–25.34 m^2^, V_n_ 1.50–15.38 m^3^.

From Table 2, the relative growth relationship, α, between the shrub and nabkha continued to decrease as the white thorn shrub grew. In the nascent stage, α values were between −0.0837 and 0.3701, and were less than the 95% confidence interval range for thickets 2 and 3, indicating that the nabkha morphology of this thicket was characterized by anisotropic growth and that the thickets grew faster than the development of the nabkha. The alpha value of the developmental stage was between 0.3163 and 0.4823, which was in the 95% confidence interval range, indicating that the morphological characteristics of the shrub nabkha at this stage were isochronous growth. The α value at the maturity stage was between 0.5334 and 0.6474, and that in thickets 7 and 8 was greater than the 95% confidence interval range, indicating that the morphological characteristics of the shrub nabkha were anisotropic growth, and the shrub growth rate was less than the development of nabkha.

### 2.2. Spatial Heterogeneity of Soil Nutrients in N. tangutorum Nabkha

As can be seen in Figure 1, with the growth in the shrub, the soil nutrients in the 0–40 cm nabkha soil significantly increased. The average SOC of 0–40 cm soil was as follows: stabilizing stage (0.28 g/kg) > developing stage (0.24 g/kg) > CK (0.24 g/kg) > rudimental stage (0.21 g/kg). TN: stabilizing stage (0.03 g/kg) > developing stage (0.02 g/kg) > CK (0.02 g/kg) > rudimental stage (0.02 g/kg). TP: stabilizing stage (0.12 g/kg) > developing stage (0.11 g/kg) > rudimental stage (0.09 g/kg) > CK (0.09 g/kg). In terms of soil depth, the nutrients in the 0–10 cm soil layer were significantly higher than those in the 10–40 cm soil layer at the rudimental stage (*p* < 0.05). There were no significant differences in the development stage, maturity stage, and CK soil layers (*p* > 0.05). With the growth of *N. tangutorum*, its nutrient accumulation ability increased. The mean SOC RII of 0–40 cm was as follows: developing stage (0.07) > developing stage (0.01) > rudimental stage (−0.07). TN RII: stabilizing stage (0.12) > developing stage (0.09) > rudimental stage (−0.05). TP RII: stabilizing stage (0.12) > developing stage (0.09) > rudimental stage (0.00). The developing stage and stabilizing stage of *N. tangutorum* showed positive effects on soil nutrient accumulation. Only TP in the rudimental stage showed nutrient accumulation.

As can be seen from Table 3, With the growth in shrub, soil C:N showed a trend of first decreasing and then increasing. The average C:N contents in the soil followed the order of CK (12.00) > rudimental stage (11.38) > stabilizing stage (10.88) > developing stage (10.06). Among them, C:N in the 10–30 cm soil layer of CK was significantly higher than that of nabkha soil at different degrees (*p* < 0.05). In terms of the nabkha soil depth at the same stage, the 10–20 cm soil layer at the developing stage was significantly lower than the other soil layers (*p* < 0.05). Average C:P contents in the soil followed the order of CK (4.42) > stabilizing stage (4.00) > rudimental stage (3.81) > developing stage (3.78). CK was significantly higher than the nabkha at the 10–20 cm soil layer (*p* < 0.05), but not at other depths. In terms of the nabkha soil depth at the same stage, the rudimental stage did not show significant differences (*p* > 0.05). The developing stage was C:P proportional to the soil depth. C:P in the stabilizing stage first increased and then decreased with the increase in soil depth. Average N:P contents in the soil followed the order of developing stage (0.22) > CK (0.21) > stabilizing stage (0.21) > rudimental stage (0.19). The difference was not significant among 0–10, 10–20, 20–30, and 30–40 cm soil depths throughout the growth stage (*p* > 0.05).

### 2.3. Morphological Characteristics of Adventitious Root System of N. tangutorum Nabkha

Figure 2 combines the growth relationship of shrub and nabkha with the change in the architecture of adventitious roots. The growth law of adventitious roots of *N. tangutorum* at different growth stages can be more clearly divided. In Figure 2A, the scatter diagram shows a clear partition. This indicates that BD is closely related to the growth stage of shrub. BD increased gradually with the growth of shrubs. The average value of BD_3_ in shrub was as follows: stabilizing stage (2.33–2.64 cm) > developing stage (1.76–1.88 cm) > rudimental stage (1.61–1.75 cm). BD_2_: stabilizing stage (1.78–1.49 cm) > developing stage (0.64–1.01 cm) > rudimental stage (0.54–1.04 cm). BD_1_: stabilizing stage (0.69–1.37 cm) > developing stage (0.39–0.72 cm) > rudimental stage (0.26–0.73 cm). In Figure 2B, the scatter plot does not show significant partitioning. This indicates that the RBD is not significantly correlated with the growth stage of the thicket (*p* > 0.05). The RBD (1:2) is more dispersed relative to the RBD (2:3), indicating that the branching pattern becomes less controllable as the adventitious roots continue to branch off. In Figure 2C,D, the scatter diagram shows a clear partition. SBR and topological indices are closely related to the shrub growth stage. The SBR increases with the growth in shrub, and the topological indices are contrary. The branches of adventitious roots gradually increase with the growth of shrubs. The branches of adventitious roots are more concentrated in SBR (2:3), and the branching rate decreased with the continuous grading of adventitious roots. After grading, the relationship between the branching rate and the development stages of the shrub gradually weakened, and the SBR (1:2) of the development stage was lower than that of the rudimental stage. The OBRr order was as follows: stabilizing stage (1.03–1.07) > developing stage (0.64–0.90) > rudimental stage (0.64–0.81). SBR (2:3) was as follows: stabilizing stage (1.50–1.88) > developing stage (1.63–1.75) > rudimental stage (0.63–1.25). SBR (1:2) was as follows: stabilizing stage (0.60–0.85) > rudimental stage (0.10–0.60) > developing stage (0.08–0.36). The root architecture of *N. tangutorum* at different growth stages was herringbone branching with a simple branch structure. However, the root structure gradually became more complex as the shrub grew. T1 was as follows: rudimental stage (0.97–0.99) > developing stage (0.86–0.88) > stabilizing stage (0.84–0.86). Qa was as follows: rudimental stage (0.93–0.96) > developing stage (0.67–0.72) > stabilizing stage (0.62–0.67). The Qb order was as follows: stabilizing stage (1.01–1.37) > developing stage (0.63–1.11) > rudimental stage (0.48–1.13).

The investigation found that adventitious roots appeared at depths ranging from 7.42 cm to 10.57 cm from the surface of the nabkha (Figure 3). Burial depth showed insignificant differences among the growth stages. The adventitious root of *N. tangutorum* was mainly vertical root, and the rudimental stages, developing stages, and stabilizing stages were 48.88%, 50.00%, and 58.33%, respectively. The inclined roots were followed by 33.34%, 37.50%, and 37.50%, respectively. The horizontal roots were the lowest (17.78%, 12.50%, 4.17%, respectively).

### 2.4. Response Mechanism of Adventitious Root Architecture of N. tangutorum to Soil Nutrients

The RDA results are analyzed in Figure 4, where the length of the lines indicates the magnitude of the overall influence and the angles between all lines indicate the strength of the correlation between the two. The first (64.45%) and second (10.98%) sorting axes of RDA cumulatively explain 75.43% of the degree of influence of soil nutrient indicators on indeterminate root configuration. BD, OBRr, Qb, and RBD of the indeterminate root configuration were clustered with soil SOC, TN, and TP on both sides of quadrants 2 and 3, and Qa and T1 were distributed at the junction of quadrant 4 and quadrant 1, so these parameters were mainly revealed by the first sorting axis. These parameters showed obvious positive correlations with each other, except for Qa and T1, which showed obvious negative correlations with these parameters. N:P and C:P were distributed in the same quadrant (2) as BD and OBRr, indicating positive correlations between them. N:P and C:P were somewhat associated with Qb, RBD, Qa, and T1, but the correlations were not close. The line of C:N was very short, indicating that it had no effect on indeterminate root configuration. C:N showed a positive correlation with Qa and T1 and a negative correlation with BD, OBRr, Qb, and RBD. As can be seen from Table 4, the importance of each soil nutrient index on the parameters of adventitious root conformation in white spurge were ranked as SOC > C:P > TP > N:P > C:N > TN. Meanwhile, SOC, C:P, and TP were the main factors influencing adventitious root system conformation. The contributions were 71.7%, 10.4%, and 6.5%, respectively.

## 3. Discussion

### 3.1. Growth Relationship between N. tangutorum Shrub and Nabkha in West Ordos

The formation of nabkha is affected by wind, sand source, and vegetation type, where vegetation type is the key factor [8,9]. For *N. tangutorum*, due to the prostrate growth and increased branching, the nabkha preferentially develops to the horizontal scale after intercepting the wind drift sand flow. After that, the vertical direction of the nabkha also increases. Finally, oval or shield-shaped nabkhas are formed [17]. In this experiment, it was found that as the shrub grew, the nabkha morphology developed gradually faster than the shrub. The growth rate of the shrub was higher than that of the nabkha at the rudimental stage. This is because the crown width in the rudimental stage is small, and the wind-breaking and sand-fixing function is weak, so the nabkha grows slowly. With the growth in *N. tangutorum*, the crown width is enlarged. The wind-breaking and sand-fixing function is improved, and the rate of nabkha formation accelerates. On the other hand, nabkha morphology development is closely related to the ability of the shrub to produce adventitious roots. Shrubs that have the ability to form adventitious roots are more likely to form larger nabkhas [10]. In this experiment, a large number of adventitious roots were found on the inner sand-buried branches after discarding the nabkha, and the adventitious root configuration showed significant changes with the growth of the shrub. Adventitious roots emerge through sand burial and act on nabkhas to expand their development and form a reciprocal feed-forward relationship. Eventually, at the stabilizing stage, the growth rate of the nabkha exceeds the growth rate of the shrub.

### 3.2. Spatial Heterogeneity of Soil Nutrients in N. tangutorum Nabkha

Affected by the moisture and temperature in the climate, the nutrient sequestration potential of the ecosystem in arid and semi-arid regions is much lower than that in other regions [18]. However, shrub cover could change the spatial distribution of soil resources and effectively improve soil nutrient storage. This principle can be used in the restoration and control process of shifting sandy land. The purpose of the fixation of shifting sand can be achieved by promoting the growth and development of shrubs [19]. In this experiment, it was found that the nutrient accumulation ability of *N. tangutorum* in the rudimental stage was negative. After that, the SOC, TN, and TP contents of nabkha gradually increased with the growth in shrub. This phenomenon is caused by the growth characteristics of the shrub itself. The rudimental stage crown width is small and cannot effectively intercept the nutrient-rich litter and fine particles. In addition, the frequent sand-drift activity is not conducive to nutrient accumulation at the early stage of nabkha formation. Therefore, the consumption of nutrients at the rudimental stage is greater than the accumulation [20]. In the developing and stabilizing stages, the shrub showed a larger crown width, and the wind-breaking and sand-fixing ability was enhanced. Litter and fine particles are deposited beneath the thicket, ultimately adding nutrients to the sandpile. Shrub cover can avoid direct sunlight and provide a better microenvironment for itself and other organisms to survive. Therefore, this promotes the growth of the shrub itself and increases the biodiversity inside the nabkha. A large amount of litter and the excreta of organisms in nabkha are produced by the growth and development of shrub. These litters release soil nutrients when they decompose and promote the mineralization of the organic matter contained in the soil, increasing soil fertility and carbon sequestration potential [21,22]. On the other hand, soil organic carbon content and stability are closely related to soil particles. With a high sand content, stable aggregates cannot easily form, and thus, soil organic matter cannot be effectively protected, resulting in the reduced carbon sequestration capacity of soil [23]. When shrub cover changes the surface flow field, the sediment content of fine particles in nabkhas increases, the proportion of stable organic carbon in sand grains increases, and the final carbon sequestration potential increases.

The nutrient content in the 0–10 cm surface layer was higher than that in the 10–40 cm soil layer. The lower precipitation in arid areas results in nutrients not being washed or leached into the lower soil layers. Ultimately, the nutrient content of the surface soil is higher than that of deeper soils. On the other hand, the increase in surface nutrient content was due to the fact that shrub cover increased litter on the surface of nabkha and increased the nutrient content of surface soil. There was no obvious nutrient stratification at the developing and stabilizing stages. Previous studies have found that rhizodegradation [24,25], an important part of plant litter decomposition, brings more SOC and TN to the soil than the above-ground part of plants, and is an important source of soil nutrients [26,27,28]. In this experiment, it was found that adventitious roots mostly grew at a depth of 7.42–10.57 cm from the surface of the nabkha. Therefore, this result may be because a number of adventitious roots grow after being buried by sand and input nutrients into the soil below 10 cm through rhizodegradation and root decomposition [29]. The adventitious roots are more developed in the developing and stabilizing stages, and therefore provide higher levels of nutrients.

In this study, the average values of C:N, C:P, and N:P in 0–40 cm soil were 10.06–11.38, 3.78–4.00, and 0.19–0.22, respectively. These are far lower than the average level of China (12.30, 52.64, 4.20) [30]. Soil C:N can characterize the decomposition rate of organic matter using soil microorganisms. C:N is less than the national average level, indicating that SOC is more deficient than TN in nabkhas, and the decomposition rate of SOC is greater than the accumulation rate [31]. This result may be due to the strong wind erosion and limited soil litter input in the West Ordos National Nature Reserve. The arid climate in the study area affects the soil’s water retention status, influences the microbial activity in the soil, accelerates the rate of SOC decomposition, and promotes SOC loss [32]. This is detrimental to the nutrient accumulation in the nabkha. The C:P of the nabkha is much lower than 200 and significantly lower than that of CK. The C:P of the nabkha is significantly lower than that of CK. This indicates that shrubs increase the mineralization rate of P and improve the availability of soil P. This also indicates that microorganisms are less limited by soil P in the process of the decomposition of organic matter [33]. N:P characterizes soil N saturation. In this study, N:P was relatively small, indicating that the limiting effect of N was greater than P during the growth in *N. tangutorum*. This may be due to the origin of N from litter and rhizodegradation. This may be due to the fact that N originates from the action of microbial communities in the soil and that the climatic environment of the arid zone affects the mineralization of nitrogen. P is derived from soil parent material, so N:P is low [20]. The stoichiometric ratios of different growth stages were relatively stable. At the same time, there was no significant difference with the change in soil depth. It may be that nabkha soil and shrub form a stable state in the long-term supply-and-demand relationship [34].

### 3.3. Response Mechanism of Adventitious Root Architecture to Soil Nutrients in Nabkha in Western Ordos

Differences in plant root architecture directly affect roots’ ability to acquire soil resources and the consumption and allocation of carbon [35,36]. It was found that the root diameter of the third-order adventitious roots directly grown on the branches buried in sand was the largest. The adventitious roots of the second-order and first-order, which branched with third-order adventitious roots, gradually tapered with the increase in diameter class. In addition, root diameters at all levels thickened with the growth in shrubs. The adventitious roots at stabilizing stages were significantly larger than those at other growth stages. This conclusion also shows again that root architecture changes with the soil environment and has plasticity [37]. This phenomenon may be due to the positive correlation between root diameter and its lifetime. A coarser root diameter can prolong its life, but its formation and death also require more carbon. Therefore, it is difficult to form a thick root diameter under a stress environment [38]. It was found that, at the early stage of nabkha formation, because of the low soil nutrient content, the root diameter of adventitious roots of the nabkha was relatively small. With the shrub growth, the soil was improved. With the increase in soil organic carbon content, the shrub gradually gained the ability to adapt to the harsh environment, and coarse adventitious roots with a longer life were selected.

Plants adjust their root architecture with continuous changes in the environment to achieve the maximum utilization of resources [39]. Previous studies have shown that plant root branching rate is negatively correlated with the degree of environmental stress [13]. This is consistent with the experimental conclusion. The branching rate of adventitious roots increased with the increase in soil nutrients, but the architecture of adventitious roots in different growth stages was herringbone branching with a simple branching structure. This is because, under the same amount of carbon input, although the distribution range of herringbone branching is smaller, it is not conducive to nutrient occupancy. However, the simple branch structure has a stronger ability to absorb and utilize resources and is more adaptable to the stress environment [15,40]. This conclusion is different from the research conclusion of Shan Lishan et al. [41] on underground root systems. The reason for this difference may be related to the difference in resources in different regions, or the difference in the characteristics of adventitious roots and underground roots. This result needs further study. No fourth-order roots were found in the adventitious root system of *N. tangutorum*. This result may be due to the inability to achieve more branching levels under a stress environment or, on the other hand, may be related to soil bulk density. Adventitious roots grow in deep soil with the development of shrub. However, the higher compactness of deep soil causes resistance to root growth and is not conducive to root expansion. Therefore, adventitious root growth and branching ability are affected [42]. In addition, RDA analysis showed that the growth in *N. tangutorum* was mainly limited by SOC and TP. The content of SOC and TP directly affected the adventitious root architecture. In addition, *N. tangutorum* can adapt to the soil environment by adjusting the root architecture, forming a reciprocal feed-forward relationship. To better adapt *N. tangutorum* to the harsh environment of West Ordos National Nature Reserve, the SOC and TP of nabkha soil can be adjusted to promote the change in adventitious root system architecture and improve the shrub’s adaptability in a stress environment.

## 4. Materials and Methods

### 4.1. Study Area

West Ordos National Nature Reserve (106°44′ E–107°43′ E, 39°13′ N–40°10′ N), located at the eastern edge of the Asian–African deserts, is a transition region between the desertification steppe of the West Ordos and the steppe desert of the East Alxa (Figure 5). Using the climate and typical vegetation under this hydrothermal condition as an indicator, the study area was classified as a desert steppe bioclimatic zone [43]. The study area displayed a large temperature difference between day and night, a dry climate, long sunshine time, strong solar radiation, and high levels of wind and sand. The annual average temperature is 7.8–8.1 °C, the annual precipitation is 162–172 mm, which mainly occurs from June to August, and the annual potential evaporation is 2470–3481 mm. The study area is dominated by a northwest wind, with average and maximum speeds of 3.2 and 24.2 m/s, respectively. The area is characterized by brown–grey calcareous soils with a high fraction of sand-sized particles and low nutrient and organic content [44]. The study area is a desertified steppe. Most of the water needed by plants comes from natural precipitation, and the groundwater depth is greater than 15 m [41]. The research site is dominated by the shrub species *N. tangutorum,* which often forms phytogenic nabkhas with a vegetation coverage of approximately 40–70%. After field investigation, it was found that adventitious roots were born on branches buried by sand. As the *N. tangutorum* grows, the adventitious roots at different growth stages show some differences in conformation.

### 4.2. Experimental Design and Methods

To ensure vigorous shrub growth in the study area, the research group went to the research area in mid-August 2021 to conduct the test. A 100 m × 100 m quadrat was set in the study area. The east–west direction was the X-axis, the north–south direction was the Y-axis, the southwest corner was the origin, and the diagonal line from the southwest corner to the northeast corner was divided into three 20 m × 20 m quadrates at equal distances. The shrub was divided into three stages: the rudimental stage, developing stage, and mature stage, based on the method of “space instead of time”, combined with the relevant literature and the actual situation of the sample plot [45]. One shrub in each of the three stages was selected for each quadrat, and a total of nine shrubs were selected. The width of shrubs (W_s_), length of shrubs (L_s_), height of shrubs (H_s_), width of nabkhas (W_n_), length of nabkhas (L_n_), and height of nabkhas (H_n_) were measured and recorded using a tape measure (Table 1).

The method of Strahler et al. [46,47] was used to grade the indefinite roots. From the outside to the inside, the outermost root was a 1st-order root, two 1st-order roots meeting this formed a 2nd-order root, and so on. If the root system met at different levels, we used the more advanced one (Figure 6 and Figure 7). The nabkha was dug with a shovel, adventitious roots were dug out, and a ruler was used to record the depth of the sand in which adventitious roots grow. The angle between the maximum order root and the horizon was measured with a protractor. This was used to represent the spatial distribution characteristics of indefinite roots (angle β < 30° is horizontal root; 60° > β > 30° is inclined root; β > 60° is vertical root). Eight 3rd-order adventitious roots and all secondary adventitious roots growing under each shrub were selected. Each root diameter (BD) was measured with vernier calipers, and the number of roots at each root level was recorded. The results were averaged to reduce the error. Considering the differences in soil nutrients caused by shrub individuals and slope direction of the nabkha, soil extraction was performed in the middle of the east-, south-, west- and north-facing slopes of nabkhas. Soil samples of 0–10 cm, 10–20 cm, 20–30 cm, and 30–40 cm were taken, totaling 16 soil samples. Soil samples of the same depth were mixed, resulting in 4 representative soil samples per shrub. Soil samples of 0–10 cm, 10–20 cm, 20–30 cm, and 30–40 cm of the same growth stage in the three sample squares were mixed, totaling 12 representative mixed soil samples. The open area away from the nabkhas and surrounded by no plant growth around it was selected to taken soil samples of 0–10 cm, 10–20 cm, 20–30 cm, and 30–40 cm in the shape of an “S”, 5 times. Soil samples of the same depth were mixed, resulting in 4 representative soil samples as CK. The nutrient test was repeated 3 times per soil sample.

### 4.3. Parameter Calculation

#### 4.3.1. Morphological Characteristic Parameters of Nabkha

The canopy (C) was calculated using the following formula [48]:(1)$$C=\pi \times {\left(\frac{L_{s}+W_{s}}{4}\right)}^{2}$$
because the nabkha is oval. Therefore, its base area (S_n_) and volume (V_n_) [49] were calculated as follows:(2)$$S_{n}=1/4\times \pi \times L_{n}\times W_{n}$$
(3)$$V_{n}=1/6\times \pi \times L_{n}\times W_{n}\times H_{n}$$

#### 4.3.2. Adventitious Root Branching Rate

Total root branching rate (OBRr): The number of roots N_i_ of each grade (i) was calculated according to Strahler’s [46,47] branch level, grade i and log_10_N_i_ were respectively used as horizontal and vertical coordinates for the plot, and the inverse logarithm of the slope of the regression line was the total root branching rate. Stepwise branching rate (SBR (i: i + 1)) was the ratio of the number of branches of level I to the number of branches of the previous level, namely:(4)$$SBR(i:i+1)=N_{i}/(N_{i}+1)$$

#### 4.3.3. Root Diameter Ratio of Adventitious Roots

Root diameter ratio (RBD (i: i+1)) is the ratio of the root of level I to the root of the previous level:(5)$$RBD(i:i+1)=BD_{i}/BD_{i+1}$$
where BD_i_ and BD_i+1_ are the diameters of grade i and grade i+1 branches, respectively.

#### 4.3.4. Topological Index

Fitter et al. [18] and Bouma et al. [21] proposed two extreme modes of root branch, herringbone branching and dichotomous branching (Figure 8), and put forward the following topological index expression:(6)T1=lgA/lgM
where T1 is the topological index, A is the total number of internal connections in the longest root channel, and M is the total number of all external root connections. The closer T1 is to 1, the closer the A and M values are, the fewer root branches there are, and the closer the shape is a to herringbone branching structure. The closer T1 is to 0.5, the closer the root structure is to dichotomous branching. Oppelt et al. [23] further studied the topology and exponential architecture and proposed a new topological index. Its expression is as follows:(7)$$Qa=\frac{a-1{-\mathrm{lbv}}_{0}}{v_{0}-1{-\mathrm{lbv}}_{0}}$$
(8)$$\mathrm{Qb}=\frac{{b-1-\mathrm{lbv}}_{0}}{(v_{0}+1)/2-1/v_{0}-lbv_{0}}$$
where Qa and Qb represent the modified values of A and B, respectively; A is equal to A in Fitter’s model; B is the average topology length, b = Pe /v_0_; Pe is the total number of all connections at the end of the base with root; lbv_0_ = lnv_0_ /ln2, where v_0_ is the same as M in Fitter’s model. The herringbone branching is Qa = Qb = 1, and the dichotomous branching is Qa = Qb =0.

#### 4.3.5. Allometric Mechanism Model

The composite factor (CH_s_/2) of shrub crown amplitude (C) and plant height (H_s_) can reflect the total basal area projected by the shrub and the volume of the above-ground part of the shrub [50]. Therefore, the relative growth relationship model was used to conduct a correlation growth analysis of CH_s_/2 and V_n_ [51], and the model form was as follows:(9)$$Y=\beta X^{\alpha }$$

After logarithmic transformation, this becomes:(10)$$\log_{10}Y=\log_{10}\beta +\alpha \log_{10}X$$
where Y is CH_s_/2, X is V_n_, β is a constant, and α is the allometric scaling exponent. When α = 1, CH_s_/2 and V_n_ grow at the same speed. When α ≠ 1, allometric growth occurs [52]; α > 1 indicates that the growth rate of the shrub is faster, while α < 1 indicates that nabkha deposition is faster. To determine whether α is equal to 1, we determined whether the 95% confidence interval of α contains 1.

#### 4.3.6. Soil Nutrient Accumulation Characteristics Calculation

The relative interaction intensity (RII: relative interaction intensity) [53] was used to indicate the nutrient enrichment of the nabkha soil.
(11)$$RII=(X_{n}-X_{i})/(X_{n}+X_{i})$$
where X_n_ and X_i_ denote the values of soil nutrient content inside and outside the nabkha, respectively. When RII > 0, it means that the nutrient content of the thicket is enriched and the “fertilizer island effect” is formed; when RII < 0, it means that the nutrient content of the thicket is reduced and the “fertilizer island effect” is not formed. The farther the RII is from 0, the stronger the effect.

### 4.4. Sample Determination

The soil samples were retrieved and air-dried in the laboratory. Experimental methods referred to Soil Agrochemical Analysis [54]. Soil organic carbon (SOC) was determined by the K_2_Cr_2_O_7_-H_2_SO_4_ oxidation method. Total nitrogen (TN) was determined by the Kjeldahl acid-digestion method. Total phosphorus (TP) was determined by the H_2_SO_4_-H_2_O_2_ digestion method.

### 4.5. Statistical Analysis

Data were preprocessed using Microsoft Excel 2010 (Microsoft Corp., Washington, DC, USA). The parameters of adventitious root configuration and soil nutrients at different growth stages were analyzed using an analysis of variance (ANOVA). The homogeneity of variances was assessed by Levine’s tests. One-sample Kolmogorov–Smirnov tests were used to validate the normality of the data distribution. All statistical analyses were completed in SPSS statistical version 25 (International Business Machines Corporation, Armonk, NY, USA) to identify significant differences at the 0.05 level. Redundancy analysis (RDA) is a sorting method combining multi-response variable regression analysis and principal component analysis (PCA). It can clearly explore the relationship between the response variable matrix and the explanatory variable matrix, and visually display this in a low-dimensional visual orthogonal sorting axis space [55]. In this study, the response variable matrix was composed of six parameters of *N. tangutorum*’s adventitious root configuration, and the explanatory variable matrix was composed of six soil nutrient-influencing factors. RDA was performed using Canoco 5.0 (Biometry, Wageningen, The Netherlands), and sequencing plots were drawn. Therefore, the correlation between soil nutrient factors in the explanatory variable matrix and adventitious root configuration parameter factors in the response variable matrix, and their influence degree, can be quantitatively analyzed so that the response law of adventitious root growth to soil nutrients can be explained more clearly. Data plots were plotted by Origin 2021 (Origin Lab, Northampton, MA, USA).

## 5. Conclusions

With the growth of shrub, the ability to accumulate sand into dunes increased, the accumulation capacity of soil nutrients increases, the root diameter at all levels gradually increased, the shrub branching gradually became more complex, and the resource utilization range gradually expanded. However, the adventitious root configuration of *N. tangutorum* at different growth stages appeared in the form of herringbone branching with a simple branching structure. The root system of *N. tangutorum* is mainly composed of vertical roots, and the depth of burial is not related to the growth stage of the shrub. SOC, C:P, and TP were the main factors affecting the configuration of the adventitious root system. Soil SOC, TP, and TN showed significant positive correlations with BD, OBRr, Qb, and RBD of the adventitious root configuration, along with significant negative correlations with Qa and T1. Meanwhile, C:P and N:P showed positive correlations with BD and OBRr, and weak correlations with Qb, RBD, Qa, and T1. C:N had a lesser effect on adventitious roots.

## Figures and Tables

**Figure 1 plants-11-03218-f001:**
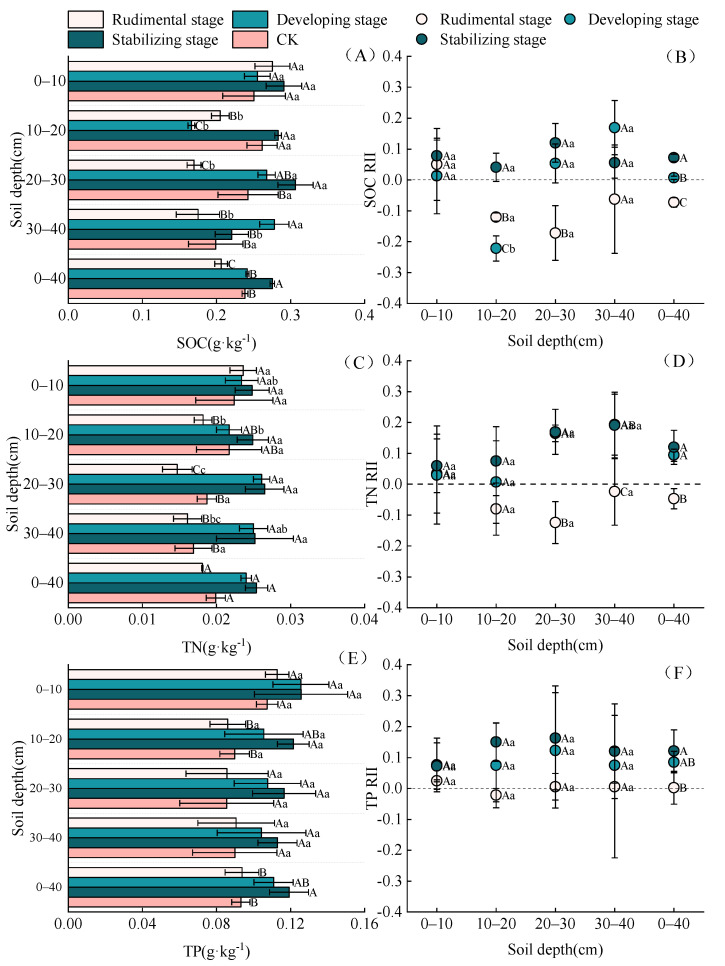
Soil nutrient content in nabkha. Capital letters indicate significant differences among different types of soil at the same depth (*p* < 0.05). Lowercase letters indicate significant differences among soils of the same type at different depths (*p* < 0.05). CK from an open area away from the nabkhas and surrounded by no plant growth. The same is given below. (**A**) Soil organic carbon (SOC), (**B**) soil organic carbon relative interaction intensity (SOC RII), (**C**) total nitrogen (TN), (**D**) total nitrogen relative interaction intensity (TN RII), (**E**) total phosphorus (TP), and (**F**) total phosphorus relative interaction intensity (TP RII).

**Figure 2 plants-11-03218-f002:**
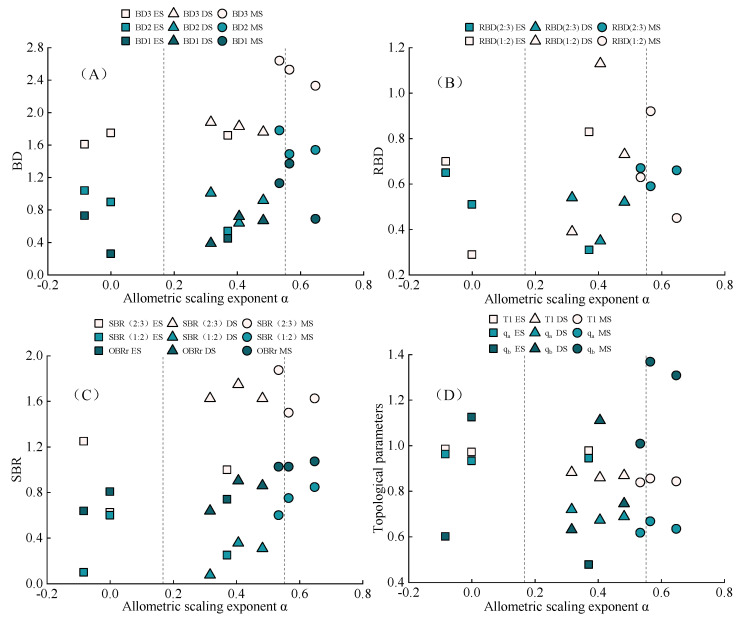
Scatter plots were created using allometric scaling exponent α as the X-axis and adventitious root architecture parameters as the Y-axis. The two reference lines indicate the growth in shrub and nabkha at the same rate. The left side of the reference line indicates that the nabkha grows faster than the shrub. The right side of the reference line indicates that the nabkha grows slower than the shrub. RS: rudimental stage; DS: developing stage; SS: stabilizing stage. (**A**) BD; (**B**) RBD; (**C**) SBR; (**D**) topological index.

**Figure 3 plants-11-03218-f003:**
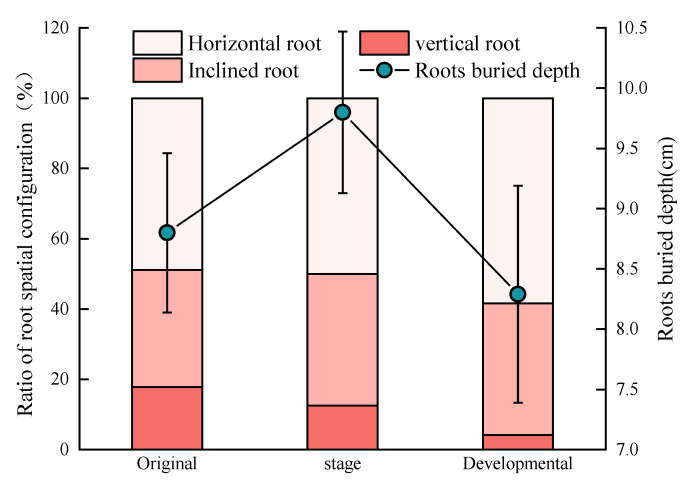
Spatial distribution characteristics of adventitious roots in *N. tangutorum* shrub.

**Figure 4 plants-11-03218-f004:**
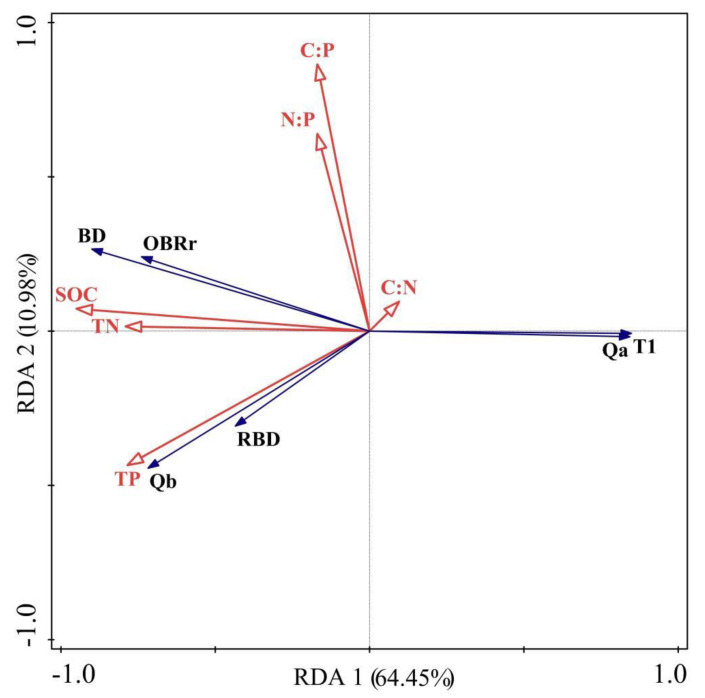
Redundancy analysis result of relationships between morphological characteristics of adventitious root system and soil stoichiometric characteristic factors.

**Figure 5 plants-11-03218-f005:**
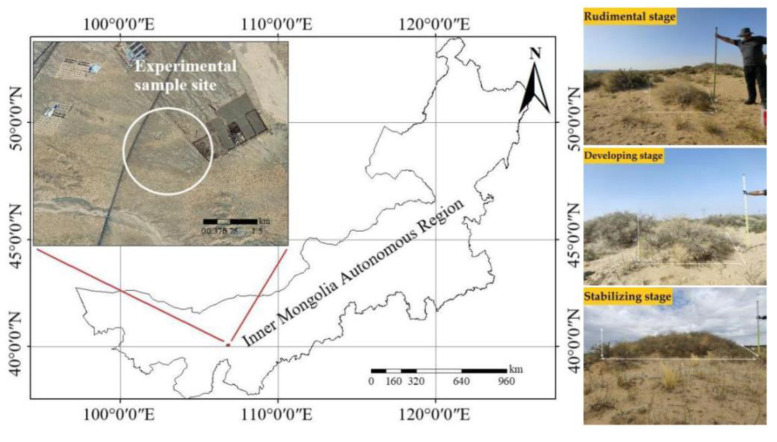
*N. tangutorum* in West Ordos Nature Reserve.

**Figure 6 plants-11-03218-f006:**
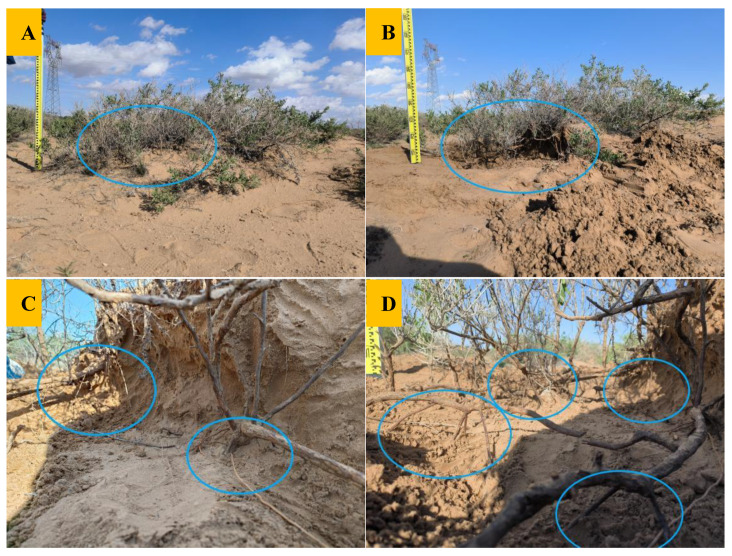
Adventitious roots inside the nabkha. (**A**) Before excavation of the nabkha; (**B**) after excavation of the nabkha; (**C**,**D**) close-up photo of adventitious roots.

**Figure 7 plants-11-03218-f007:**
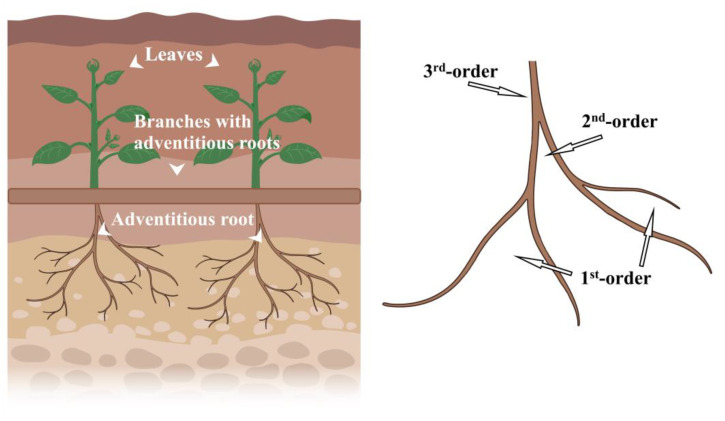
Grading method of adventitious roots.

**Figure 8 plants-11-03218-f008:**
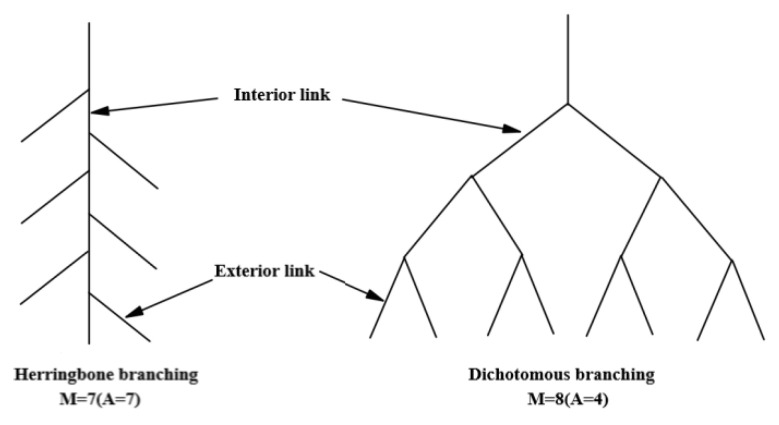
Schematic diagram of root topological structure.

**Table 1 plants-11-03218-t001:** Morphological characteristics of shrub and nabkha. Width of shrubs (W_s_), length of shrubs (L_s_), height of shrubs (H_s_), canopy (C), width of nabkhas (W_n_), length of nabkhas (L_n_), height of nabkha (H_n_), base area of nabkhas (S_n_), and volume of nabkhas (V_n_).

Growth Stage		W_s_ (m)	L_s_ (m)	H_s_ (m)	C (m^2^)	W_n_ (m)	L_n_ (m)	H_n_ (m)	S_n_ (m^2^)	V_n_ (m^3^)
Rudimental stage	1	2.41	3.19	0.99	3.43	2.61	3.23	0.49	6.62	2.16
	2	2.32	2.87	0.91	2.81	2.28	2.74	0.46	4.91	1.50
	3	2.34	2.94	0.86	2.84	2.31	2.99	0.48	5.42	1.74
Developing stage	4	3.78	4.15	0.99	5.19	3.76	4.18	0.68	12.34	5.60
	5	3.32	3.94	1.09	4.97	3.56	3.98	0.64	11.13	4.75
	6	3.24	3.27	1.06	3.68	3.36	3.88	0.56	10.24	3.82
Stabilizing stage	7	4.32	5.14	1.33	8.22	5.06	5.88	0.84	23.37	13.09
	8	4.32	5.50	1.54	9.73	5.23	6.17	0.91	25.34	15.38
	9	4.07	4.76	1.41	7.47	5.11	5.99	0.89	24.04	14.26

**Table 2 plants-11-03218-t002:** The allometric relationships and coefficient by log_10_CH_s_/2 = log_10_β + αlog_10_V_d_ for the morphological characteristics of *N. tangutorum* and nabkha.

Growth Stage	Allometric Scaling Exponent			95% Confidence Interval	Allometric Relationships
	Log_10_CH_s_/2	Log_10_V_n_	α		
Rudimental stage	0.2300	0.3350	0.3701	0.1671–0.5519	Isokinetic
	0.1060	0.2428	−0.0010		Allometric
	0.0861	0.2395	−0.0837		Allometric
Developing stage	0.4096	0.7479	0.4055		Isokinetic
	0.4326	0.6765	0.4823		Isokinetic
	0.2903	0.5824	0.3163		Isokinetic
Stabilizing stage	0.7377	1.1168	0.5653		Allometric
	0.8747	1.1868	0.6474		Allometric
	0.7218	1.1542	0.5334		Isokinetic

**Table 3 plants-11-03218-t003:** Stoichiometric ratio of soil organic carbon, total nitrogen, and total phosphorus at different growth stages.

Stoichiometric Ratio	Soil Depth (cm)	Growth Stage			
		CK	Rudimental Stage	Developing Stage	Stabilizing Stage
C:N	0–10	19.52 ± 1.44 Aa	20.29 ± 3.08 Aa	18.85 ± 1.31 Aa	20.22 ± 0.36 Aa
	10–20	21.25 ± 3.55 Aa	19.46 ± 1.13 Aa	13.26 ± 1.45 Bb	19.69 ± 1.68 Aa
	20–30	22.21 ± 2.19 Aa	20.11 ± 1.79 ABa	17.69 ± 0.80 Ba	20.09 ± 2.24 ABa
	30–40	20.55 ± 4.94 Aa	19.14 ± 5.03 Aa	19.24 ± 1.40 Aa	15.36 ± 1.84 Ab
	0–40	20.68 ± 1.51 A	19.61 ± 0.67 AB	17.34 ± 0.44 C	18.76 ± 0.87 BC
C:P	0–10	4.02 ± 0.62 Aab	4.21 ± 0.35 Aa	3.52 ± 0.31 Aab	4.06 ± 0.48 Aab
	10–20	5.02 ± 0.08 Aa	4.12 ± 0.23 Ba	2.78 ± 0.53 Cb	4.03 ± 0.23 Bab
	20–30	5.07 ± 0.96 Aa	3.60 ± 1.11 Aa	4.36 ± 0.71 Aab	4.58 ± 0.59 Aa
	30–40	3.86 ± 0.27 Ab	3.37 ± 0.41 Aa	4.79 ± 1.32 Aa	3.41 ± 0.63 Ab
	0–40	4.42 ± 0.15 A	3.81 ± 0.31 B	3.78 ± 0.37 B	4.00 ± 0.33 AB
N:P	0–10	0.21 ± 0.05 Aa	0.21 ± 0.03 Aa	0.19 ± 0.01 Aa	0.20 ± 0.02 Aa
	10–20	0.24 ± 0.04 Aa	0.21 ± 0.02 Aa	0.21 ± 0.04 Aa	0.21 ± 0.03 Aa
	20–30	0.23 ± 0.07 Aa	0.18 ± 0.06 Aa	0.25 ± 0.05 Aa	0.23 ± 0.01 Aa
	30–40	0.20 ± 0.05 Aa	0.19 ± 0.06 Aa	0.25 ± 0.09 Aa	0.23 ± 0.06 Aa
	0–40	0.21 ± 0.02 A	0.19 ± 0.02 A	0.22 ± 0.03 A	0.21 ± 0.01 A

**Table 4 plants-11-03218-t004:** Significance order and significance level test of soil organic carbon (SOC), total nitrogen (TN), total phosphorus (TP), and soil stoichiometric characteristic factors.

Name	Explains %	Contribution %	Pseudo-F	p	Importance Sequencing
SOC	58.4	71.7	9.8	0.006	1
C:P	8.5	10.4	1.5	0.206	2
TP	5.3	6.5	1.0	0.416	3
N:P	3.9	4.7	0.5	0.648	4
C:N	3.0	3.7	0.3	0.748	5
TN	2.3	2.9	0.4	0.782	6

## Data Availability

All the data supporting the conclusions of this article are included in this article.

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
