# Peer review of "Response Mechanisms of Adventitious Root Architectural Characteristics of Nitraria tangutorum Shrubs to Soil Nutrients in Nabkha"

_plants, 2022, doi:10.3390/plants11233218_

Round 1
Reviewer 1 Report
This is an interesting manuscript on the edaphic evolution and structure forming Nitraria tangutorum Bobrov, Sovetsk. However, there are several methodological errors, as well as typing formatting errors.
On the other hand, the composition of Nitraria tangutorum is like that of other shrubs such as Ziziphus lotus L. or Maytenus senegalensis subsp. europaea Boiss Rivas Mart. ex Güemes & M.B. Crespo, which are extreme phreatophyte shrubs of arid zones and sandy soils, which, due to the accumulation of sand, form false dunes. In this type of vegetation, nutrient cycles and growth are conditioned not so much by the availability of water as by the seasonal xericity of the climate. (Thomas, F. M. 2014. Ecology of phreatophytes. In Progress in botany (pp. 335-375). Springer, Berlin, Heidelberg). Therefore, without a solid bioclimatic characterisation, it is not possible to adequately contextualise the study or extrapolate the results and methodologies to other vegetation types with similar physiognomy.
Comments can be found in the attached file

Author Response
Dear Editor,
Thank you for your email and the reviewer’s comments concerning our manuscript entitled“Response mechanism of adventitious root architecture characteristics of Nitraria tangutorum shrub to soil nutrients in nabkha”(ID:plants-1904169). We thank the reviewer for the very helpful comments.We revised the manuscript according to the recommendations. The modifications are highlighted in red. We provide below with a brief“List of changes”in the revised manuscript, followed by our answers to the referee’s comments.
List of changes:
- We touched up the English writing of the manuscript. Revised formatting errors that appeared in the manuscript.
- Unnecessary keywords were removed.
- Additional literature was introduced in appropriate places.
- The introduction was revised to make its purpose clearer.
- The study methodology was described in more detail and the rationale for the choice of this approach was explained.
- Fig. 1 has been revised to make the study area clearer.
- Fig. 2 and Fig. 3 have been added to more clearly represent the study design.
- The title was changed to "Mechanism of response of adventitious root architecture characteristics of Nitraria tangutorumshrubs to soil nutrients in nabkha".
Detalled reply to reviewer’s comments:
Comments 1:Please, the first time a scientific name appears, attach the authorship, as reflected in the International Code of Botanical Nomenclature.
Response to comment 1:The article has been amended in accordance with the International Code of Botanical Nomenclature.
Comments 2:Please check the formatting of the text. The dots are not correctly formatted.
Response to comment 1:Thank you for your careful examination and the format of the manuscript has been corrected.
Comments 3:First of all, check the formatting of the text, numbers are unnecessary. Secondly, try not to use the same words in the title as keywords.
Response to comment 3:The format has been modified. Delete the keyword "Nitraria tangutorum" after consideration.
Comments 4:More bibliography is needed. The starting hypothesis is not clear. The main and secondary objectives of the study are not clear and concise.
Response to comment 4:The introduction to the manuscript has been revised and relevant references have been added where necessary. For details, see the revised red-letter section in the article.
Comments 5:Please, explain the meaning of "fat island"
Response to comment 5:Thank you for your comment, here is the wording is not accurate. The section has been reworked and the description has been added to the manuscript. The details are as follows: However, scrub’s influence, through interception, root secretion, and apoplankton, creates a "fertilizer island effect", in which the nutrients of nabkha soils are higher than those without vegetation cover
Comments 6:This paragraph needs to be better written, part of it would be better in Materials and Methods.
Response to comment 6:This paragraph has been rewritten, and part of the content is given over to materials and methods. For details, see the revised red-letter section in the article.
Comments 7:It would be much more illustrative to attach a map with the location to contextualise the environmental characteristics of the study area.
Response to comment 7:I have revised Fig. 1. The new overview map of the study area shows the location more clearly.
Comments 8:A homogeneous bioclimatic classification of the territory is required (see for instance Rivas-Martínez, 2011)
Response to comment 8:After reviewing relevant literature, the manuscript was revised, and the discussion part included the discussion on bioclimatic characteristics. The modifications are as follows: West Ordos National Nature Reserve (106°44'E-- 107°43'E; 39°13'N- 40°10'N), located at the eastern edge of the Asian–African deserts, is a transition region between the desertification steppe of the west Ordos and the steppe desert of the East Alxa. Using the climate and typical vegetation under this hydrothermal condition as an indicator, the study area was classified as desert steppe bioclimatic zones [17]. The study area displayed a large temperature difference between day and night, a dry climate, long sunshine time, strong solar radiation and high levels of wind and sand. The annual average temperature is 7.8-8.1 ℃, the annual precipitation is 162-172mm, mainly from June to August, and the annual potential evaporation is 2470-3481mm.
Comments 9:The statistical section is too poor and inaccurate. The previous analysis of the data to decide which statistical test to use is omitted or not done. How do they know that they have to use an ANOVA test (which is a parametric test that assumes the precepts of Normal distribution and Heterocedasticity)?
Response to comment 9:The data analysis section was rewritten. Explain why one-way analysis of variance was chosen and what problem was chosen to solve. The specific modifications are as follows: Analysis of variance (ANOVA) was used to test the significance of the differences between the means of two or more samples. In this study, the parameters of adventitious root configuration and soil nutrients at different growth stages were analyzed with ANOVA. This was used to compare whether there was a significant difference in the variation in parameters caused by shrub growth. ANOVA with IBM SPSS statistical version 25 (International Business Machines Corporation, Armonk, NY, USA) showed significant differences at the 0.05 level.
Comments 10:Why do you use redundancy analysis, and not other multivariate methods such as Principal Component Analysis, or other types of multiple regression?
Response to comment 10:Redundancy analysis is a sorting method combining multi-response variable regression analysis and principal component analysis (PCA). It can clearly explore the relationship between the response variable matrix and explanatory variable matrix, and visually display this in a low-dimensional visual orthogonal sorting axis space. In this study, the response variable matrix was composed of six parameters of N. tangutorum's adventitious roots’ configuration, and the explanatory variable matrix was composed of six soil-nutrient-influencing factors. RDA was performed using Canoco 5.0 (Biometry, Wageningen, Netherlands) and sequencing plots were drawn. Therefore, the correlation between soil nutrient factors in the explanatory variable matrix and adventitious root configuration parameter factors in the response variable matrix, and their influence degree can be quantitatively analyzed so that the response law of adventitious root growth to soil nutrients can be explained more clearly. Data plots were plotted by Origin 2021 (Origin Lab, Northampton, MA, USA).
Comments 11:How have you come to know which variables are involved, without a prior principal factor analysis?
Response to comment 11:Thank you for your comments on my manuscript. Regarding the selection of variables involved, our research group found that the growth of shrub roots was mainly influenced by SOC, TN and TP after reading a lot of literature. After several group discussions and experiments. This experiment finally determined the selection of SOC, TN, TP and related stoichiometric parameters as variables. Therefore, this experiment was based on the previous studies to further investigate the relationship between the change of scrub root configuration and soil nutrients.
Reference:
[1] Fitter A H , Stickland T R . Architectural analysis of plant root systems 2. Influence of nutrient supply on architecture in contrasting plant species[J]. New Phytologist, 1991,118(2):383-389.
[2] Fitter A H. The topology and geometry of plant-root systems - influence of watering rate on root-system topology in trifolium-pratense[J]. Annals of Botany, 1986,58(1):91-101.
[3] Fitter A H, et al. Architectural analysis of plant root systems : III. Studies on plants under field conditions[J]. New Phytologist, 1992,121(2):243-248.
[4] Liu L B,Zhong Q L,Ni J.Ecosystem C:N:P stoichiometry and storages of a secondary plateau-surface karst forest in Guizhou Province,southwestern China. Acta Ecologica Sinica,2019,39(22):8606-8614.
[5] Ladanai S., Agren G.I., Olsson B.A. Relationships Between Tree and Soil Properties in Picea abies and Pinus sylvestris Forests in Sweden[J]. Ecosystems, 2010, 13(2):302-316.
[6] Wu J., Sheng M.Y., Xiao H.L., Guo C., Wang L.J. Fine root architecture of adaptive plants and its correlation with nutrient stoichiometric characteristics of fine root and rhizosphere soils in karst rocky desertification environments, SW China. Acta Ecologica Sinica,2022,42(2):677-687.
[7] Zhao Y Y, Lu Z H, Xia J B, Liu J T. Root architecture and adaptive strategy of 3 shrubs in Shell Bay in Yellow River Delta. Acta Ecologica Sinica, 2015, 35(6):1688-1695.
[8] Ma X Z, Wang X P. Root architecture and adaptive strategy of two desert plants in the Alxa Plateau. Acta Ecologica Sinica, 2020, 40(17):6001-6008.
Comments 12:Without a clear and concise methodological design in the statistical section, the results may not be reliable, and may not be adjusted to the evolution of the phenomenon to be studied.
Because the statistical analysis is not well structured, the results may not be reliable. Therefore, a discussion of the results could lead the study to wrong conclusions.
Response to comment 12:A change was made to the research methodology section, a change that explains why this research methodology was chosen and what phenomena it was ultimately chosen to explain. Also, Figure 2 and Figure 3 were added to more visually represent the study design. The revisions are marked in red in the manuscript.
Comments 13:Please explain briefly the content of the table.
Response to comment 13:The table illustrates the morphological characteristics of scrub and nabhas. Its purpose is to provide an account of the background of the study. These morphological parameters are briefly described in the paragraph 3.1.
Comments 14:Please check the formatting of the bibliography, words in bold, phrases in italics, etc.
Response to comment 14:Thank you for your careful review, the reference format has been checked and revised.
Comments 15:The composition of Nitraria tangutorum is like that of other shrubs such as Ziziphus lotus L. or Maytenus senegalensis subsp. europaea Boiss Rivas Mart. ex Güemes & M.B. Crespo, which are extreme phreatophyte shrubs of arid zones and sandy soils, which, due to the accumulation of sand, form false dunes. In this type of vegetation, nutrient cycles and growth are conditioned not so much by the availability of water as by the seasonal xericity of the climate. (Thomas, F. M. 2014. Ecology of phreatophytes. In Progress in botany (pp. 335-375). Springer, Berlin, Heidelberg). Therefore, without a solid bioclimatic characterisation, it is not possible to adequately contextualise the study or extrapolate the results and methodologies to other vegetation types with similar physiognomy.
Response to comment 15:Thank you for your comments, which provided a very good idea for my manuscript. After literature review and field investigation, my academic research group found that the formation of nabkhas is based on the growth of plants and adventitious roots. The nabkhas is affected by the "fertilizer island effect", and the "fertilizer island effect" has different degrees of effect depending on the growth of the shrub. The shrub constantly adjusts its root system configuration to adapt to the soil nutrient changes. Therefore, this experiment is intended to reveal the changes in the "fertilizer island effect" due to the growth of the scrub. Under the influence of the "fertilizer island effect", the root system configuration responds to changes in the soil environment and the main influencing factors are identified.
Therefore, this experiment focused on the association between soil nutrients and root conformation aspects during scrub growth in the same bioclimatic zone. However, knowledge about bioclimates was added in the revision of the manuscript to make the article more broadly applicable.
We thank the reviewer and remain at your disposal for any futher questions.
Yours sincerely,
Xiao-le Li

Reviewer 2 Report
Dear Authors,
I have reviewed the introduction and methods sections of your article; unfortunately, I have needed to stop my review before entering the Results, Discussion and Abstract in fine detail.
I have three major concerns:
1. The introduction does not adequately provide the required background information; primary references (e.g. Grime) and concepts on ecological stress are not well defined. This concept of ecological stress is potentially contentious and should either be bolstered with more information and associated references or completely removed entirely. For example, the authors might approach the topic along an ecological continuum concept (e.g. different strategies appear along different positions of an ecological gradient) rather than stress per se. I like the study area and idea; it is a nice area of research. However, I think the introduction would be better served (especially to a broader audience) if a larger angle related to plant clonality is integrated.
2. The methods are descriptive; however, I am a visual creature because I could not visualise the study design or how the calculations were made. For this reason, I suggest that the methods are not reproducible. This is particularly true; formulas 6 and 10 require ‘log’ transformations (there are many different bases to choose from, so I can not reproduce those formulas). I suggest the authors make supplementary descriptive material showing hypothetical values and how each measure is calculated (nothing can be lost in translation).
3. I found several sentences and spelling errors (see minor comments); please use Grammarly and make sure that it is up to standard.
Minor comments:
L37: please remove Nitraria from keywords (it is already in the title and adds nothing, choose another word, e.g. clonality).
L47: correct “years.Previous”, no space.
L47–51: Place in the broader context of plant clonality
L52: Missing article
L53–56: Sentence with three different topics; please split and simplify.
L58: Missing introduction of ecological stress (and citations from P. Grime) – this is already a contentious issue – please see my comments above.
L67: I do not know what ‘improvement’ means.
L74: correct “stages.The”
L76: “stress environment” see above (L58), also consider changing to something less contentious, e.g. ‘ecological gradient’.
L77: ghost ‘.’ at the end of N. tangutorum
L81: Missing article; the entire line is not a sentence.
L84–85: Missing citation
L90: missing article and fullstop.
L92–93: please expand the ecological significance of the sentence/stress/gradient.
L95: What test?
L95–105: Please draw the study area with a diagram and all axes’ labels.
L96: please correct. Do the authors mean “We selected XX patches of nabka located within carefully selected 100 * 100 m quadrat missing any signs of external disturbance”…?
L99 & 102: correct spacing
L103–104: what do the d and g stand for?
L109–110: it makes no sense.
L110: Ghost capital letter.
L111: should read “It was used”
L117: please replace ~ with the appropriate en-type dash meaning from A to B (e.g. A–B) and not the estimation symbol.
L121–122: The figure needs some landmarks (e.g. major towns, ca. 2/3). Where does the elevation data source come from? The figure caption should be expanded as A and B and redone. There is a map and some pictures of the study stages (actually, I would remove them since there is a person in the picture, perhaps they could serve better in the appendix).
L125: Should there be ‘were’ since many canopy covers are calculated?
L127: correct spacing
L166: Specify which type of log transformation.
L179–183: Not enough detail. Please tell us what the authors are explicitly testing. Further, stick to SPSS since it offers pre and host-hock testing (and testing assumptions), which have not been explained. Please tell us what matrices have gone into the RDA analysis in Canoco. Canoco is missing a citation.
Author Response
Dear Editor,
Thank you for your email and the reviewer’s comments concerning our manuscript entitled“Response mechanism of adventitious root architecture characteristics of Nitraria tangutorum shrub to soil nutrients in nabkha”(ID:plants-1904169). We thank the reviewer for the very helpful comments.We revised the manuscript according to the recommendations. The modifications are highlighted in red. We provide below with a brief “List of changes” in the revised manuscript, followed by our answers to the referee’s comments.
List of changes:
- We touched up the English writing of the manuscript. Revised formatting errors that appeared in the manuscript.
- Unnecessary keywords were removed.
- Additional literature was introduced in appropriate places.
- The introduction was revised to make its purpose clearer.
- The study methodology was described in more detail and the rationale for the choice of this approach was explained.
- Fig. 1 has been revised to make the study area clearer.
- Fig. 2 and Fig. 3 have been added to more clearly represent the study design.
- The title was changed to "Mechanism of response of adventitious root architecture characteristics of Nitraria tangutorumshrubs to soil nutrients in nabkha".
Detalled reply to reviewer’s comments:
Comments 1:The introduction does not adequately provide the required background information; primary references (e.g. Grime) and concepts on ecological stress are not well defined. This concept of ecological stress is potentially contentious and should either be bolstered with more information and associated references or completely removed entirely. For example, the authors might approach the topic along an ecological continuum concept (e.g. different strategies appear along different positions of an ecological gradient) rather than stress per se. I like the study area and idea; it is a nice area of research. However, I think the introduction would be better served (especially to a broader audience) if a larger angle related to plant clonality is integrated.
Response to comment 1:The introductory section of the manuscript has been rewritten. This revision provides a detailed description of the concept of nomenclature, while introducing additional literature to support it. The detailed revisions have been highlighted in red in the original text. The details are as follows: Nabkha is a geomorphological form, formed by the accumulation of wind and sand flow encountering scrub blockage in arid, semi-arid and semi-humid desert areas [1]. Nabkha contains a large amount of information on regional environmental changes, which can comprehensively reflect the dynamic environment and material sources in the deposition area, and provide an important parameter indicator to describe the regional soil wind erosion and land degradation status [2]. However, scrub’s influence, through interception, root secretion, and apoplankton, creates a "fertilizer island effect", in which the nutrients of nabkha soils are higher than those without vegetation cover [3]. This can better promote plant growth to achieve the effect of fixing drifting sand [4]. Therefore, scrub has been a hot research topic in recent years. The formation of nabkhas depends on multiple influences, such as sand source, wind and plants, among which vegetation is a key factor affecting the morphology of nabkhas [5]. It is generally agreed that the plant wind-blocking benefits depend on the plant sidelight area on the windward side [6-7], and taller and denser shrubs are more likely to form nabkhas due to their stronger interception capacity [8-9]. However, some of the scrubs cannot form larger nabkhas although they possess the ability to accumulate sand into dunes. It has been shown that the formation of nabkhas is strongly related to their adventitious roots, and plants that can produce adventitious roots after being buried in sand are more likely to form tall nabkhas [10]. Therefore, it is important to explore adventitious roots to further understand nabkhas.
The root system is an important organ for material exchanges between plants and the external environment [11]. Root system conformation is the distribution and arrangement of the root system in the soil [12]. To adapt to the environment to the greatest extent possible, the root architecture of plants dynamically adjusts to changes in the soil environment [13]. A good root architecture facilitates plants’ ability to improve the root’s morphological stability and resource utilization efficiency during adversity stress [11]. This is a feedback mechanism and part of the plants’ ecological adaptation strategy to heterogeneous soil resources [14]. It has been found that different plants exhibit different root adaptation strategies in similar environments [15]. Plants at different growth stages adapt to changing environments by adjusting their root configuration [16]. What kind of adaptation strategies provide an adventitious root conformation under the influence of the nabkha soil environment exhibit?
There is a lack of research on the adventitious roots’ strategies to adapt to stressful environments during the growth in desert scrub. It is also unclear how the nabkha soil environment affects adventitious root configuration. In view of this, this experiment was conducted to investigate the characteristics of adventitious root conformation and the nutrients of the nabkha soil at different growth stages. The aim was to reveal the ecological strategy of the adventitious root system of white spurge scrub in adapting to stressful environments, reveal the response mechanism of the adventitious root system configuration to the soil environment and to identify the key factors affecting this, and provide a theoretical basis and data support for the conservation and restoration of desert plants.
Comments 2:The methods are descriptive; however, I am a visual creature because I could not visualise the study design or how the calculations were made. For this reason, I suggest that the methods are not reproducible. This is particularly true; formulas 6 and 10 require ‘log’ transformations (there are many different bases to choose from, so I can not reproduce those formulas). I suggest the authors make supplementary descriptive material showing hypothetical values and how each measure is calculated (nothing can be lost in translation).
Response to comment 2:To show my research design more visually. I added more photos in this revision (Figure 2 and Figure 3). Comparative photos of the scrub before and after it was dug up and close up pictures of the adventitious roots within the nabkha are shown in Figure 2.The grading of adventitious roots is shown more clearly by Figure 3.
In Equation 6, A is the total number of internal connections of the longest root channel; M is the total number of all external root connections. Figure 4 is used to explain the location of A and M. Both A and M in this experiment were recorded during the field survey (data from eight adventitious roots selected for each thicket were recorded and eventually averaged). Table 1 was used to demonstrate these calculations.
The calculation results of Equation 10 have been put into Table 2. Making linear equation of lgCH and lgVd. Equation y=2.0463x - 0.5315 is obtained, where -0.5315 is lgβ.
Table 1. Topology parameters
|
A |
M |
lgA |
lgM |
T1 |
1 |
8.18 |
8.57 |
0.91 |
0.93 |
0.98 |
2 |
7.21 |
7.63 |
0.86 |
0.88 |
0.97 |
3 |
8.73 |
9.01 |
0.94 |
0.95 |
0.99 |
4 |
7.63 |
10.64 |
0.88 |
1.03 |
0.86 |
5 |
8.32 |
11.44 |
0.92 |
1.06 |
0.87 |
6 |
8.09 |
10.69 |
0.91 |
1.03 |
0.88 |
7 |
7.41 |
10.39 |
0.87 |
1.02 |
0.86 |
8 |
7.91 |
11.63 |
0.90 |
1.07 |
0.84 |
9 |
8.56 |
12.95 |
0.93 |
1.11 |
0.84 |
Table 2.The allomoteric relationships and coefficient by lgCH=lgβ+αlgVd for morphological characteristics of N. tangutorum and nabkha
|
CH |
lgCH |
lgvd |
lgβ |
α |
1 |
2.772 |
0.442793226 |
0.334453751 |
-0.5315 |
2.9131 |
2 |
2.366 |
0.37401474 |
0.176091259 |
-0.5315 |
5.1423 |
3 |
2.2704 |
0.356102378 |
0.240549248 |
-0.5315 |
3.6899 |
4 |
3.9303 |
0.594425701 |
0.748188027 |
-0.5315 |
1.5049 |
5 |
3.9567 |
0.597333123 |
0.67669361 |
-0.5315 |
1.6682 |
6 |
3.4556 |
0.538523465 |
0.582063363 |
-0.5315 |
1.8383 |
7 |
6.2909 |
0.798712782 |
1.116939647 |
-0.5315 |
1.1909 |
8 |
7.5614 |
0.878602213 |
1.186956335 |
-0.5315 |
1.1880 |
9 |
6.2322 |
0.794641382 |
1.154119526 |
-0.5315 |
1.1491 |
Comments 3:I found several sentences and spelling errors (see minor comments); please use Grammarly and make sure that it is up to standard.
Response to comment 3:Thank you for your careful review of my manuscript. I have carefully checked the manuscript in its entirety and corrected any errors that occurred.
We thank the reviewer and remain at your disposal for any futher questions.
Yours sincerely,
Xiao-le Li

Round 2
Reviewer 1 Report
Everything is now much clearer and better written. The manuscript is easy to follow. Congratulations to the authors.
However there are still some issues that would improve the understanding of the manuscript.
Introduction:
It would be very appropriate to include the following text in the main and secondary objectives section. This text, comes from the response the authors sent to this reviewer. "Therefore, this experiment aims to reveal the changes in the "fertiliser island effect" due to scrub growth. Under the influence of the "fertiliser island effect", the configuration of the root system responds to changes in the soil environment and the main influencing factors are identified.
"This experiment focused on the association between soil nutrients and aspects of root conformation during shrub growth in the same bioclimatic zone."
Material and Methods:
Any and all figures, formulae and tables, must be cited in the text. In addition, it is recommended that they should be arranged close to the text they are intended to illustrate.
- Line 86: Remove the extra hyphen, and specify the coordinate (Long 106°44'E - 107°43'E; Lat 39°12'N - 40°10'N) not as it is in the text (106°44'E-- 107°43'E; 39°13'N- 40°10'N).
- Line 112: Why did you do this survey in mid-August?
- Line 150: In the first review, I didn't notice, but the canopy is the area of the tree canopy that projects onto the ground. It is therefore a surface area whose SI units are m2. However, your formula (1) would give units of dimension 1 (metres), not dimension 2 (square metres). Please review and reflect on the formula.
- Formula (6): Express the base of the logarithm.
- Line 209: If you use a parametric ANOVA analysis to analyse the differences between two samples, your data have to meet the requirements of normal distribution and homoscedasticity that this statistical tool demands. Without a prior analysis with a normality test such as Saphiro-Wilks, you cannot know whether you can correctly use an ANOVA test or its non-parametric equivalent the Kruskal-Wallis test.
Results:
- Line 248: Figure 3 or figure 5?
- Line 266: If you talk about "significant values" always add the p-value.
- Line 280-283: Please conform to the text size format for figure captions as explained in the journal.
- Line 346-352: The variable TP, according to your figure 8 (which by the way is not cited in the text) correlates best with RBD and Qb, I think it is a typing error. Perhaps the authors were referring to the variable TN which, according to their figure 8, is correlated with BD and OBRr. On the other hand the correlation between C:P and N:P with BD and OBRr are quite doubtful according to the graph, which shows some negative correlation with Qb and RBD.
Discussion:
4.1. In this section, the architecture of the adventitious roots found in their experiment is not provided. These results are not really discussed. Please restructure this section accordingly.
Line 365-366; This sentence is not understood.
Line 411-412: In this type of vegetation, there is a phenomenon typical of arid sites, and that is that because there is very little precipitation, nutrients cannot be washed or leached down to the lower layers. Please discuss this fact, with the result that the authors have obtained that the nutrient content in the surface layers is higher than in the deeper layers.
Lines 424-441: Carbon and nitrogen are the only two nutrients whose cycles are mixed (atmospheric and soil). While plants take up carbon from the atmosphere directly, nitrogen must be taken up by the soil microbiota. In this context, in semi-arid sites, organic matter takes a long time to decompose, while nitrification is much more constant (Catalán, N., Marcé, R., Kothawala, D. N., & Tranvik, L. (2016). Organic carbon decomposition rates controlled by water retention time across inland waters. Nature Geoscience, 9(7), 501-504). Please discuss this fact with the results obtained in this section.
Conclusions
Please include a section in which you express the conclusions of your research. Remember that conclusions must be inferred from the results obtained.
Author Response
Dear Editor:
Thank you for your email and the reviewer’s comments concerning our manuscript entitled“Response mechanism of adventitious root architecture characteristics of Nitraria tangutorum shrub to soil nutrients in nabkha”(ID:plants-1904169). We thank the reviewer for the very helpful comments.We revised the manuscript according to the recommendations. The modifications are highlighted in red. We provide below with a brief“List of changes”in the revised manuscript, followed by our answers to the referee’s comments.
List of changes:
- Question on the language regarding the manuscript. I contacted the translation agency of the MDPI to revise my manuscript.
- The title of the manuscript was modified to "Response mechanisms of adventitious root architectural characteristics of Nitraria tangutorumshrubs to soil nutrients in nabkha"
- I rewrote the abstract section. Research topics and research methods are highlighted in the previous section. A large number of descriptions of the results were removed, making the most important results clearer. For the results section, which content should be deleted and which content should be retained, I refer to the way other papers were written.
- The introduction section was modified, and the purpose and meaning became clearer.
- The structure of the text is adjusted according to the journal requirements, and the method and purpose are put in Part 4.
- The figures and tables in the results were modified with reference to the newly published papers in this journal. The title includes a more detailed description of the pictures and tables, the abbreviations are explained in more detail, and such modifications are more appropriate.
- Picture colors appearing in the manuscript were replaced. Pictures that fit the same characteristics are used in similar colors (Figure 1). At the same time, the patterns and colors in the picture were also adjusted (Figure 2). This change made the manuscript more uniform in color and a clearer presentation of the results.
- The discussion section has been extensively modified, see details below.
- A new conclusion section was added.
Detalled reply to reviewer’s comments:
Comments 1:It would be very appropriate to include the following text in the main and secondary objectives section. This text, comes from the response the authors sent to this reviewer. "Therefore, this experiment aims to reveal the changes in the "fertiliser island effect" due to scrub growth. Under the influence of the "fertiliser island effect", the configuration of the root system responds to changes in the soil environment and the main influencing factors are identified.
Response to comment 1:Changes have been made to the manuscript based on your comments, as follows: Therefore, our experiment aimed to reveal the changes in the "fertilizer island effect" caused by shrub growth in the same bioclimatic zone. In this study, the response of the root morphology to changes in the soil environment under the influence of the "fertilizer island effect" and the main influencing factors are identified
Comments 2:Any and all figures, formulae and tables, must be cited in the text. In addition, it is recommended that they should be arranged close to the text they are intended to illustrate.
Response to comment 2:The content of this section has been revised and the corresponding citation locations have been indicated in the text.
Comments 3:Line 86: Remove the extra hyphen, and specify the coordinate (Long 106°44'E - 107°43'E; Lat 39°12'N - 40°10'N) not as it is in the text (106°44'E-- 107°43'E; 39°13'N- 40°10'N).
Response to comment 3:Revisions have been made to this section.
Comments 4:Line 112: Why did you do this survey in mid-August?
Response to comment 4:There are two reasons for this. One is to ensure that the plants in the study area are in a vigorous state of growth, and the other is because the group has other activities that need to take up time. Ultimately after discussion it was decided to use this time as the experiment time. A reasonable explanation has been added in the corresponding place in the manuscript.
Comments 5:Line 150: In the first review, I didn't notice, but the canopy is the area of the tree canopy that projects onto the ground. It is therefore a surface area whose SI units are m2. However, your formula (1) would give units of dimension 1 (metres), not dimension 2 (square metres). Please review and reflect on the formula.
Response to comment 5:Thank you for your careful examination of my manuscript. This part was an error due to my oversight, and I have rechecked and recalculated that part. The formulae are cited with references to ensure correct use.
Comments 6:Line 209: If you use a parametric ANOVA analysis to analyse the differences between two samples, your data have to meet the requirements of normal distribution and homoscedasticity that this statistical tool demands. Without a prior analysis with a normality test such as Saphiro-Wilks, you cannot know whether you can correctly use an ANOVA test or its non-parametric equivalent the Kruskal-Wallis test.
Response to comment 6:Changes have been made to this section as follows: The parameters of adventitious root configuration and soil nutrients at different growth stages were analyzed using an analysis of variance (ANOVA). The homogeneity of variances was assessed by Levine’s tests. One-sample Kolmogorov–Smirnov tests were used to validate the normality of the data distribution. All statistical analyses were completed in SPSS statistical version 25 (International Business Machines Corporation, Armonk, NY, USA) to identify significant differences at the 0.05 level.
Comments 7:If you talk about "significant values" always add the p-value.
Response to comment 7:Changes have been made to this section as follows: The homogeneity of variances was assessed by Levine’s tests. One-sample Kolmogorov–Smirnov tests were used to validate the normality of the data distribution. All statistical analyses were completed in SPSS statistical version 25 (International Business Machines Corporation, Armonk, NY, USA) to identify significant differences at the 0.05 level.
Comments 8:Line 346-352: The variable TP, according to your figure 8 (which by the way is not cited in the text) correlates best with RBD and Qb, I think it is a typing error. Perhaps the authors were referring to the variable TN which, according to their figure 8, is correlated with BD and OBRr. On the other hand the correlation between C:P and N:P with BD and OBRr are quite doubtful according to the graph, which shows some negative correlation with Qb and RBD.
Response to comment 8:Changes have been made to this section as follows: The RDA results are analyzed in Figure 4, where the length of the lines indicates the magnitude of the overall influence and the angles between all lines indicate the strength of the correlation between the two. The first (64.45%) and second (10.98%) sorting axes of RDA cumulatively explain 75.43% of the degree of influence of soil nutrient indicators on indeterminate root configuration. BD, OBRr, Qb and RBD of the indeterminate root configuration were clustered with soil SOC, TN and TP on both sides of quadrants 2 and 3, and Qa and T1 were distributed at the junction of quadrant 4 and quadrant 1, so these parameters were mainly revealed by the first sorting axis. These parameters showed obvious positive correlations with each other, except for Qa and T1, which showed obvious negative correlations with these parameters. N:P and C:P were distributed in the same quadrant (2) as BD and OBRr, indicating positive correlations between them. N:P and C:P were somewhat associated with Qb, RBD, Qa and T1, but the correlations were not close. The line of C:N was very short, indicating that it had no effect on indeterminate root configuration. C:N showed a positive correlation with Qa and T1, and a negative correlation with BD, OBRr, Qb and RBD.
Comments 9:4.1. In this section, the architecture of the adventitious roots found in their experiment is not provided. These results are not really discussed. Please restructure this section accordingly.
Response to comment 9:This paragraph cites extensive literature demonstrating the influence of scrub on sandpile formation and also shows the important role that adventitious roots play in the formation of sandpiles. The results of this experiment corroborate this relative growth relationship. A description of the changes in adventitious roots during the growth of the scrub is also included. Although this paragraph does not provide a direct description of the adventitious root configuration, this can be used as a background for the introduction and in combination with 3.3 to better highlight the necessity of adventitious roots in the sandpile formation process.Therefore I think this paragraph is very necessary. However there are still some writing problems.
Thanks to your suggestions, I have reworked this section as follows: The formation of nabkha is affected by wind, sand source and vegetation type, where vegetation type is the key factor [8–9]. For N. tangutorum, due to the prostrate growth and increased branching, the nabkha preferentially develops to the horizontal scale after intercepting the wind drift sand flow. After that, the vertical direction of the nabkha also increases. Finally, oval or shield-shaped nabkhas are formed [17]. In this experiment, it was found that as the shrub grew, the nabkha morphology developed gradually faster than the shrub. The growth rate of the shrub was higher than that of the nabkha at the rudimental stage. This is because the crown width in the rudimental stage is small, and the wind-breaking and sand-fixing function is weak, so the nabkha grows slowly. With the growth in N. tangutorum, the crown width is enlarged. The wind-breaking and sand-fixing function is improved, and the rate of nabkha formation accelerates. On the other hand, nabkha morphology development is closely related to the ability of the shrub to produce adventitious roots. Shrubs that have the ability to form adventitious roots are more likely to form larger nabkhas [10]. In this experiment, a large number of adventitious roots were found on the inner sand-buried branches after discarding the nabkha, and the adventitious root configuration showed significant changes with the growth of the shrub. Adventitious roots emerge through sand burial and act on nabkhas to expand their development and form a reciprocal feed-forward relationship. Eventually, at the stabilizing stage, the growth rate of the nabkha exceeds the growth rate of the shrub.
Comments 10:Line 411-412: In this type of vegetation, there is a phenomenon typical of arid sites, and that is that because there is very little precipitation, nutrients cannot be washed or leached down to the lower layers. Please discuss this fact, with the result that the authors have obtained that the nutrient content in the surface layers is higher than in the deeper layers.
Response to comment 10:The section has been revised as follows: The lower precipitation in arid areas results in nutrients not being washed or leached into the lower soil layers. Ultimately, the nutrient content of the surface soil is higher than that of deeper soils.
Comments 11:Lines 424-441: Carbon and nitrogen are the only two nutrients whose cycles are mixed (atmospheric and soil). While plants take up carbon from the atmosphere directly, nitrogen must be taken up by the soil microbiota. In this context, in semi-arid sites, organic matter takes a long time to decompose, while nitrification is much more constant (Catalán, N., Marcé, R., Kothawala, D. N., & Tranvik, L. (2016). Organic carbon decomposition rates controlled by water retention time across inland waters. Nature Geoscience, 9(7), 501-504). Please discuss this fact with the results obtained in this section.
Response to comment 11:Thanks for your comments and providing a very good idea for my manuscript revision. I have read this reference, and have revised my manuscript. The section has been revised as follows: The arid climate in the study area affects the soil’s water retention status, influences the microbial activity in the soil, accelerates the rate of SOC decomposition and promotes SOC loss [32]. This is detrimental to the nutrient accumulation in the nabkha. The C:P of the nabkha is much lower than 200 and significantly lower than that of CK.
Comments 12:Please include a section in which you express the conclusions of your research. Remember that conclusions must be inferred from the results obtained
Response to comment 12:A conclusion section has been added, as follows: With the growth of shrub, the ability to accumulate sand into dunes increased, and the accumulation capacity of soil nutrients increases, the root diameter at all levels gradually increased, the shrub branching gradually became more complex, and the resource utilization range gradually expanded. However the adventitious root configuration of N. tangutorum at different growth stages appeared in the form of herringbone branching with a simple branching structure. The root system of N. tangutorum is mainly composed of vertical roots, and the depth of burial is not related to the growth stage of the shrub. SOC, C:P and TP were the main factors affecting the configuration of the adventitious root system. Soil SOC, TP, and TN showed significant positive correlations with BD, OBRr, Qb and RBD of the adventitious root configuration, and significant negative correlations with Qa and T1. C:P and N:P showed positive correlations with BD and OBRr, and weak correlations with Qb, RBD, Qa and T1. C:N had a lesser effect on adventitious roots.
We thank the reviewer and remain at your disposal for any futher questions.
Yours sincerely,
Xiao-le Li

Reviewer 2 Report
Dear Authors,
I have reviewed your article's abstract, introduction, methods and results sections; unfortunately, I stopped my review before entering the Discussion.
I have two major concerns:
Starting with the title (L2–4), which is incorrectly formatted and should read “Response mechanisms of adventitious root architectural characteristics of Nitraria tangutorum shrubs to soil nutrients in nabkha”. This error (and many others) highlights the need to use an external grammar editor. I have found several small mistakes in my first round and again in this revision that have hindered the flow and readability of the manuscript.
The abstract needs major attention and should be restructured. When writing the abstract, focus on the following sections: Aim, Methods, Results, Discussion/Conclusion. As I see it, they have addressed the first (AIM); however, its formulation was not correct. Methods and results have been fused. As a result, I can not tell which methods the authors have used at a glance. Use of archaic English (herringbone) leading to potential confusion from the onset should be avoided. There is one sentence putting their results in context – here, the authors have left the angle of ‘stress’ (see also L80), which appears as a lingering and potentially controversial topic identified by my previous revisions.
Please note that not all of my previous comments were addressed. Please see my new comments from this round of revision below too:
Abstract
L13–15: It is too specific and does not introduce the general aim of the paper. For example; We studied the morphology of adventitious roots belonging to Nitraria tangutorum Bobrov, Sovestsk changed along with the formation of nakha – a special type of shrubland emerging as micro-inselberg islands in a sea of mobile sand.
L27: “Herringbone” is archaic English jargon and unlikely to be understood by readers; remove it with more descriptive terminology.
L20, 22, 33, 34,35 & 36: All contain abbreviations that have not been introduced and detract from the general flow.
Introduction
The introduction is moving in the right direction, however, still needs attention from a native English speaker. Please also consider including references and work on the study of trait ontogeny.
L42: How can geomorpholgy accumulate wind?
Methods
L86: “106°44'E--” ?
L90: zone
L93–94, XXX, XXX, XXX: (en-type dash) – must be used instead it means from a to b. Check throughout for consistency.
L95: with an average and maximum speed of 3.2 and 24.2m/s, respectively.
L96–98:
Original: The main soil types are brown calcium soil and grey desert soil, which are characterized by a shallow soil layer, rough soil quality, poor nutrient content and low humus content
Problems. Many. Repetitive use of soil (5 times).
Suggestion: The areas is characterized by brown–grey calcareous soils with a high fraction of sand-sized particles and low nutrient and organic content.
L98–103: Condense to one sentence. Please be careful what you mean by ‘dominant’ and ‘accompanying’ plants versus statistical fidelity; these are all important terms with their own meaning. In any case, it is not the objective of the study to define such terms (see demystifying dominance paper), and perhaps these could all be lumped, or an appropriate reference included (e.g. a syntaxonomic paper).
L108: what is a burner?
L116: What is ‘this’. The order of this sentence and the pre/proceeding sentences should be checked as the antecedent is unclear.
L120–122: What is the axis referring too, the canopy or the roots? What does the g and d abbreviation stand for? Please note that I have asked for this clarification in my previous revision too.
L134–136: I do not understand. Do you mean that to get a representative soil sample you collected to a depth of 40 cm in four locations within the quadrat? Then you divided these collections into four depth categories and mixed them? Forming for aggregated and representative samples.
L138–140: are all redundant since you have highlighted the root measurement procedure above.
L150, 153: Formulas need to correct spacing and use of the × (multiplication symbol) and not the X (capital letter).
L158: What logarithm base? There are many and still, I do not know.
L178: What is lg? It is not explained.
L179: What is lbv (it is not explicitly linked in the description.
L196–200: In the Table 2, different classes of allometric relationships have been defined. I would like to know how you have arrived at these classes (i.e. isokinetic, allometry)
Subsection Statistical analyses
The authors have made substantial changes and explained the procedures used per my previous request in more detail. However, please consider the following simplifications:
L209–210: “Analysis of variance (ANOVA) was used to test the significance of the 210 differences between the means of two or more samples”. Can be deleted, the proceeding sentence explains clearly what means and what groups are being tested. Thanks
L212–213: Delete
L213–215: Tighten, e.g. we used IBM SPSS statistical version 25 (citation) and alpha 0.05 to perform the ANOVAs and search for differences in the variation of our parameters caused by shrub growth.
L215–228: Simplify. E.g. We performed an RDA ordination of the response and explanatory matrix and presented a two-dimensional biplot with explanatory variables fitted as vectors; these vectors were maintained if they explained a significant proportion of the compositional variance determined by post-hoc permanova analyses on the RDA using XXX permutations?.
Results
L328: There are more than one result, please fix the title to Results.
L242, 245: En-type dash. Please check throughout. Use consistent unit spacing (either with or without a space, check throughout the ms).
L247: Please explain all the letters in the table caption. I should be able to read it as a standalone text.
L248: I don’t see α or white thorn scrub in the table 2. Further and observational inferences of white thorn scrub have not been included in the methods section.
L260: Table 2 the allometric relationship classes not defined in the methods. In any case it should not read ‘allometry’ rather, allometric.
L263: What is this? Please formulate the sentence again.
L264, 273: en-type dash
L265: consistent use of superscript throughout, many instances you have used C/m or Hd/m actually that is the same as Cm-1 etc and should be changed for consistency throughout the ms.
L280: Figure 5. Use the en-type dash. Move the significance letters further from the bar axes for visibility. CK not explained in the caption. Please use consistent colouring scheme (e.g. why does the developing stage change from green to orange?). Delete: The same is given below.
L284: English.
L354: Figure 8. Are the values in brackets eigenvalues or percentages, please fix.
L356: Table 4. Names with abbreviations not explained in the table. Please also consider removing he table to save space (relegate to the appendix). Please also consider removing some of the red vectors from the RDA which are not significant players. Was the development stage included as a random or fixed effect in the permanovas? Have you tried fitting squared terms of all variables in the model and running variable reduction (this would help to determine if there are non-linear responses) too.
Author Response
Dear Editor:
Thank you for your email and the reviewer’s comments concerning our manuscript entitled“Response mechanism of adventitious root architecture characteristics of Nitraria tangutorum shrub to soil nutrients in nabkha”(ID:plants-1904169). We thank the reviewer for the very helpful comments.We revised the manuscript according to the recommendations. The modifications are highlighted in red. We provide below with a brief“List of changes”in the revised manuscript, followed by our answers to the referee’s comments.
List of changes:
- Question on the language regarding the manuscript. I contacted the translation agency of the MDPI to revise my manuscript.
- The title of the manuscript was modified to "Response mechanisms of adventitious root architectural characteristics of Nitraria tangutorumshrubs to soil nutrients in nabkha"
- I rewrote the abstract section. Research topics and research methods are highlighted in the previous section. A large number of descriptions of the results were removed, making the most important results clearer. For the results section, which content should be deleted and which content should be retained, I refer to the way other papers were written.
- The introduction section was modified, and the purpose and meaning became clearer.
- The structure of the text is adjusted according to the journal requirements, and the method and purpose are put in Part 4.
- The figures and tables in the results were modified with reference to the newly published papers in this journal. The title includes a more detailed description of the pictures and tables, the abbreviations are explained in more detail, and such modifications are more appropriate.
- Picture colors appearing in the manuscript were replaced. Pictures that fit the same characteristics are used in similar colors (Figure 1). At the same time, the patterns and colors in the picture were also adjusted (Figure 2). This change made the manuscript more uniform in color and a clearer presentation of the results.
- The discussion section has been extensively modified, see details below.
- A new conclusion section was added.
Detalled reply to reviewer’s comments:
Comments 1:Starting with the title (L2–4), which is incorrectly formatted and should read “Response mechanisms of adventitious root architectural characteristics of Nitraria tangutorum shrubs to soil nutrients in nabkha”. This error (and many others) highlights the need to use an external grammar editor. I have found several small mistakes in my first round and again in this revision that have hindered the flow and readability of the manuscript.
Response to comment 1:Thanks for your advice, since I am not a native English speaker, there are many problems in the use of English. Therefore, in order to address the language problems present in the manuscript, we have contacted MDPI's translation agency to translate my manuscript. I hope that this problem can be resolved.
Comments 2:The abstract needs major attention and should be restructured. When writing the abstract, focus on the following sections: Aim, Methods, Results, Discussion/Conclusion. As I see it, they have addressed the first (AIM); however, its formulation was not correct. Methods and results have been fused. As a result, I can not tell which methods the authors have used at a glance. Use of archaic English (herringbone) leading to potential confusion from the onset should be avoided. There is one sentence putting their results in context – here, the authors have left the angle of ‘stress’ (see also L80), which appears as a lingering and potentially controversial topic identified by my previous revisions.
Response to comment 2:I rewrote the abstract section. Research topics and research methods are highlighted in the previous section. A large number of descriptions of the results were removed, making the most important results clearer. For the results section, which content should be deleted and which content should be retained, I refer to the way other papers were written. Hopefully this revision could be with my manuscript getting better.
Comments 3:L13–15: It is too specific and does not introduce the general aim of the paper. For example; We studied the morphology of adventitious roots belonging to Nitraria tangutorum Bobrov, Sovestsk changed along with the formation of nakha – a special type of shrubland emerging as micro-inselberg islands in a sea of mobile sand.
Response to comment 3:This section has been modified, specifically as follows: The adventitious roots of desert shrubs respond to a nabkhas soil environment by adjusting their configuration characteristics, but the mechanism of this response and the main influencing factors are still unclear. To illustrate this response pattern, the Nitraria tangutorum Bobrov, Sovetsk. in West Ordos National Nature Reserve was studied, and the shrub was divided into three growth stages: the rudimental stage, developing stage and stabilizing stage. A combination of total root excavation and root tracing was used to investigate their adventitious root morphology. The results show that: (1) As the shrub grows, the ability to accumulate sand into nabkhas increases. (2) The soil nutrient accumulation capacity increased with shrub growth. The "fertilizer island effect" was formed in the nutrient developing stage and stabilizing stage of nabkhas soil, but the rudimental stage was not formed. (3) The adventitious root architecture of N. tangutorum at different growth stages was all herringbone with simple branch structure. With the growth in N. tangutorum, the root diameter of each level gradually increased, the branches of the shrub grew gradually complicated, and the range of resource utilization gradually expanded. (4) Redundancy analysis (RDA) results shows that soil organic carbon (SOC) was the main factor affecting the adventitious root architecture.
Comments 4:L27: “Herringbone” is archaic English jargon and unlikely to be understood by readers; remove it with more descriptive terminology.
Response to comment 4:Herringbone is a Proper Noun, so I didn't modify it. Therefore, instead of modifying it to any other noun, I chose to add a more descriptive explanation to the term. I hope this is a balance between accurate expression and simplicity and vividness.
Comments 5:L20, 22, 33, 34,35 & 36: All contain abbreviations that have not been introduced and detract from the general flow.
Response to comment 5:Thanks for your comment, the issue has been modified.
Comments 6:The introduction is moving in the right direction, however, still needs attention from a native English speaker. Please also consider including references and work on the study of trait ontogeny.
Response to comment 6:Thanks for your comments, I have made changes to the language issue.
Comments 7:L93–94, XXX, XXX, XXX: (en-type dash) – must be used instead it means from a to b. Check throughout for consistency.
Response to comment 7:Thanks for your comments, I have revised the use of en-type dash.
Comments 8:L98–103: Condense to one sentence. Please be careful what you mean by ‘dominant’ and ‘accompanying’ plants versus statistical fidelity; these are all important terms with their own meaning. In any case, it is not the objective of the study to define such terms (see demystifying dominance paper), and perhaps these could all be lumped, or an appropriate reference included (e.g. a syntaxonomic paper).
Response to comment 8:Thanks for your comments, I have revised it. I found a newly published article in the Plants journal, whose research area is near my study area, and our situation is very similar. I referred to its writing method and hope that this revision will solve the problem in my manuscript. Specific modifications are made as follows: The research site is dominated by the shrub species N. tangutorum, which often forms phytogenic nabkhas with a vegetation coverage of approximately 40–70%.
Reference
Bao, F.; Xin, Z.; Liu, M.; Li, J.; Gao, Y.; Lu, Q.; Wu, B. Preceding Phenological Events Rather than Climate Drive the Variations in Fruiting Phenology in the Desert Shrub Nitraria tangutorum. Plants 2022, 11, 1578. https://doi.org/ 10.3390/plants11121578
Comments 9:L120–122: What is the axis referring too, the canopy or the roots? What does the g and d abbreviation stand for? Please note that I have asked for this clarification in my previous revision too.
Response to comment 9:My statement may not accurately convey what I mean, and the axis here refers to the tree crown. The g means the shrub and the d means the nabkha. This is a common way of expression in Chinese, so I habitually choose these two letters. This is a common way of expression in Chinese, so I habitually choose these two letters. I refer to a published paper and hope to revise my manuscript.
Reference
Jasem M. Al-Awadhi • Ali M. Al-Dousari. Morphological characteristics and development of coastal nabkhas, north-east Kuwait. Int J Earth Sci (Geol Rundsch) (2013) 102:949–958.DOI 10.1007/s00531-012-0833-9
Comments 10:L134–136: I do not understand. Do you mean that to get a representative soil sample you collected to a depth of 40 cm in four locations within the quadrat? Then you divided these collections into four depth categories and mixed them? Forming for aggregated and representative samples.
Response to comment 10:My expression does not clearly express my meaning. I have modified this part, specifically as follows: Soil extraction was performed in the middle of the east-, south-, west- and north-facing slopes of nabkhas. Soil samples of 0–10 cm, 10–20 cm, 20–30 cm and 30–40 cm were taken, totaling 16 samples. Soil samples of the same depth were mixed, resulting in four representative soil samples per shrub. Soil samples with the same growth stage and the same soil depth in three small samples were mixed and divided into three replicates. Soil was taken in the same way as CK from an open area away from the nabkhas and surrounded by no plant growth.
Comments 11:L150, 153: Formulas need to correct spacing and use of the × (multiplication symbol) and not the X (capital letter).
Response to comment 11:I used × in my manuscript, not capital X. It may be due to a little error on the display of the PDF. I have corrected the erroneous use of spaces present in the manuscript, and for other formatting issues. Thanks for the careful inspection.
Comments 12:L158: What logarithm base? There are many and still, I do not know.
L178: What is lg? It is not explained.
L179: What is lbv (it is not explicitly linked in the description.
Response to comment 12:I have modified the section. The base number is 10. The implications of lbv have relevant explanations in the manuscript, and its algorithm is also described in detail, specifically as follows: lbv0 = lnv0 /ln2, where v0 is the same as M in Fitter.
Comments 13:L196–200: In the Table 2, different classes of allometric relationships have been defined. I would like to know how you have arrived at these classes (i.e. isokinetic, allometry)
Subsection Statistical analyses
The authors have made substantial changes and explained the procedures used per my previous request in more detail. However, please consider the following simplifications:
Response to comment 13:My research group has applied this method several times, and has published several papers. My research group discussed this method many times and thought it was feasible. Therefore, I continue to apply this method based on the results of previous research, and on this basis, I have reviewed the relevant literature to ensure its accuracy.
References
- JOCHEN SCHENK; ROBERT B.JACKSON. Rooting depths, lateral root spreads and below-ground/aboveground allometries of plants in water-limited ecosystems. Journal of Ecology2002, 90, 480-494.
Wei, Y.J.; Liu, M.Y.; Wang, J.; et al. The effects of vegetation communities on soil organic carbon stock in an enclosed desert-steppe region of northern China. Soil fertility 2022, 68, 284-294.
Dang, X.H.; Gao, Y.; Meng, Z.J.; et al. Biomass distribution pattern and prediction model of five desert dominant shrubs in western Ordos region. Journal of Desert Research 2017, 37, 100-108.
Jackson, S. Rooting Depths,Lateral Root Spreads and Below-Ground/Above-Ground Allometries of Plants in Water-Limited Ecosystems. Journal of Ecology 2002, 90, 480-494.
Niklas, K.J. Modelling below- and above-ground biomass for non-woody and woody plants. Annals of Botany 2005, 95, 315-321.
Sun Y, He H J, Li L, Song C M, Wang F J, Xia Fu C. Biomass allocation and biomass allometric models of six early-summer herbs under the canopy of broad-leaved Korean pine forest during different growth periods in Jiaohe, Jilin Province. Acta Ecologica Sinica 2017, 37, 6523-6533.
Comments 14:L209–210: “Analysis of variance (ANOVA) was used to test the significance of the differences between the means of two or more samples”. Can be deleted, the proceeding sentence explains clearly what means and what groups are being tested. Thanks
Response to comment 14:Modifications have been made, specifically as follows: The parameters of adventitious root configuration and soil nutrients at different growth stages were analyzed using an analysis of variance (ANOVA). The homogeneity of variances was assessed by Levine’s tests. One-sample Kolmogorov–Smirnov tests were used to validate the normality of the data distribution. All statistical analyses were completed in SPSS statistical version 25 (International Business Machines Corporation, Armonk, NY, USA) to identify significant differences at the 0.05 level.
Comments 15:L213–215: Tighten, e.g. we used IBM SPSS statistical version 25 (citation) and alpha 0.05 to perform the ANOVAs and search for differences in the variation of our parameters caused by shrub growth.
L215–228: Simplify. E.g. We performed an RDA ordination of the response and explanatory matrix and presented a two-dimensional biplot with explanatory variables fitted as vectors; these vectors were maintained if they explained a significant proportion of the compositional variance determined by post-hoc permanova analyses on the RDA using XXX permutations?.
Response to comment 15:Modifications have been made, specifically as follows: Redundancy analysis (RDA) is a sorting method combining multi-response variable regression analysis and principal component analysis (PCA). It can clearly explore the relationship between the response variable matrix and explanatory variable matrix, and visually display this in a low-dimensional visual orthogonal sorting axis space [55].
Comments 16:L242, 245: En-type dash. Please check throughout. Use consistent unit spacing (either with or without a space, check throughout the ms).
Response to comment 16:The full text has been examined and revised.
Comments 17:L247: Please explain all the letters in the table caption. I should be able to read it as a standalone text.
L248: I don’t see α or white thorn scrub in the table 2. Further and observational inferences of white thorn scrub have not been included in the methods section.
Response to comment 17:This section has been modified. To ensure the correctness of this revision, I refer to the writing format of the newly published papers in the same journal. Hope that your problem can be solved.
Comments 18:L280: Figure 5. Use the en-type dash. Move the significance letters further from the bar axes for visibility. CK not explained in the caption. Please use consistent colouring scheme (e.g. why does the developing stage change from green to orange?). Delete: The same is given below.
Response to comment 18:The implications of CK are described in detail. The picture color change was made for aesthetic reasons, and this section has now been modified. Here I would like to emphasize that Figures 1 use different colors for the purpose of distinguishing the growth stages. But in Figure 2, using the same method is not more intuitive to see my research results. So I have modified it here to choose to represent the growth stages with different patterns. In the picture, the meaning of the different colors and different patterns has been clearly described, so I think this modification is advantageous for a clearer understanding of the results.
Comments 19:L356: Table 4. Names with abbreviations not explained in the table. Please also consider removing he table to save space (relegate to the appendix). Please also consider removing some of the red vectors from the RDA which are not significant players. Was the development stage included as a random or fixed effect in the permanovas? Have you tried fitting squared terms of all variables in the model and running variable reduction (this would help to determine if there are non-linear responses) too.
Response to comment 19:Thanks for your comments, I have revised this section. My experiments selected root configuration parameters and corresponding soil indicators at different growth stages as influence factors. My research group has used this method many times, and has published several papers. For the idea you gave me, it is very novel, it gave me a very great inspiration, thanks! However, my research group is more familiar with the current approach, so I'm not going to try this great idea. Thank you once again for your thoughts.
We thank the reviewer and remain at your disposal for any futher questions.
Yours sincerely,
Xiao-le Li

Round 3
Reviewer 1 Report
Congratulations to the authors, the manuscript is now suitable for publication in this journal.
Author Response
The experience of submitting articles to your journal has left me a very good impression. Thank you for your comments on my manuscript. Your ideas and suggestions help me improve my manuscript constantly, which is very important for my manuscript writing.
I wish you success in your work, and I wish the editorial department better.